

# Tropical Dry Forest Response to Nutrient Fertilization: A Model Validation and Sensitivity Analysis

Shuyue Li[1], Bonnie G. Waring[2], Jennifer S. Powers[3,4], David Medvigy[1]

[1]Department of Biological Sciences, University of Notre Dame, Notre Dame, IN 46556 USA
[2]Grantham Institute on Climate Change and the Environment, Imperial College London, South Kensington, London, SW7 2AZ UK
[3]Ecology, Evolution and Behavior, University of Minnesota, St. Paul, Minnesota 55108 USA
[4]Department of Plant and Microbial Biology, University of Minnesota, St. Paul, Minnesota 55108 USA

*Correspondence to*: David Medvigy (dmedvigy@nd.edu)

**Abstract.** Soil nutrients, especially nitrogen (N) and phosphorus (P), regulate plant growth and hence influence carbon fluxes between the land surface and atmosphere. However, how forests adjust biomass partitioning to leaves, wood, and fine roots in response to N and/or P fertilization remains puzzling. Recent work in tropical forests suggests that trees increase fine root production under P fertilization, but it is unclear whether mechanistic models can reproduce this dynamic. In order to better

understand mechanisms governing nutrient effects on plant allocation and improve models, we used the nutrient enabled ED2 model to simulate a fertilization experiment being conducted in a secondary tropical dry forest in Costa Rica. We evaluated how different allocation parameterizations affected model performance. These parameterizations prescribed a linear relationship between relative allocation to fine roots and soil P concentrations. The slope of the linear relationship was allowed to be positive, negative, or zero. Some parameterizations realistically simulated leaf, wood and fine root production, and these

parameterizations all assumed a positive relationship between relative allocation to fine roots and soil P concentration. On a thirty-year timescale, under unfertilized conditions, our model predicted the largest aboveground biomass (AGB) accumulation when relative allocation to fine roots was positively related to soil P concentration. However, this result was mostly driven by increased water use rather than decreased nutrient limitation. On a thirty-year timescale with P fertilization, the assumption of a positive correlation between relative allocation to fine roots and soil P concentration led to over-investment to fine roots and

reductions in vegetation biomass. Our study demonstrates the need of simultaneous measurements of leaf, wood, and fine root production in nutrient fertilization experiments. Models that do not accurately represent allocation to fine roots may be highly biased in their simulations of AGB, especially when simulating a range of sites with significantly different soil P concentrations.



## 1 Introduction

Primary production in many terrestrial ecosystems is likely to be limited by nitrogen (N), phosphorus (P), or both (LeBauer and Treseder 2008; Hou et al. 2020). Because nutrient availability modulates plant growth and death, it can determine terrestrial carbon storage (Oren et al., 2001), affect tree mortality and recovery after disturbance events (Gessler et al., 2016), and even alter the sign and magnitude of land carbon sink in response to climate change (Wieder et al., 2015). Free-Air $CO_2$ Enrichment (FACE) experiments have demonstrated that insufficient N and/or P can eventually halt the initial stimulation of aboveground

growth by increased $CO_2$ (e.g. Norby et al., 2010; Reich and Hobbie, 2013; Terrer et al., 2019). Earth System Models (ESMs) also highlight the effects of nutrient limitation. The C[4]MIP-CMIP6 models that account for N cycling exhibit a 25–30% lower $CO_2$ fertilization effect on land carbon storage than models that do not (Canadell et al., 2021). However, there is significant variation across models (Arora et al., 2020), suggesting the need for increased process-level understanding.

Nutrient fertilization experiments in the field can be used to assess the effects of nutrient limitation on terrestrial ecosystems

and to improve models. In tropical moist forests, several studies found little effect of nutrient fertilization on stand-level carbon accumulation (Wright et al., 2011; Alvarez-Clare et al., 2013; Schulte-Uebbing and de Vries 2018; Wright et al. 2018; Báez and Homeier, 2018), but relatively strong effects on the productivity of particular species and size classes. In Panama, the addition of N and K was associated with increased growth rates of sapling and pole-sized trees but not of larger size classes (Wright et al., 2011). A fertilization experiment in central Amazonia reported increased NPP with P fertilization but not N

fertilization; in particular, fine root production increased under P addition but slightly decreased under N addition (Lugli et al., 2021; Cunha et al., 2022). In Costa Rica, the addition of P led to faster wood growth for smaller stems but this effect was not significant for all size classes (Alvarez-Clare et al., 2013). Basal area growth was found to vary between species depending on their traits in a nutrient fertilization experiment in an Ecuadorian tropical montane forest (Báez and Homeier, 2018). At this forest, common species with acquisitive traits (traits supporting fast resource acquisition and growth rates, including high stem

conductivity, high specific leaf area, high foliar N and P concentrations, and low wood density) generally had stronger responses to nutrient addition (mainly N+P) than trees without acquisitive traits. At the community level, nutrient fertilization did not have a statistically significant effect on aboveground biomass productivity.

Compared to tropical moist forests, fewer fertilization experiments have been done in tropical dry forests. In a Mexico, P fertilization led to strongly increased basal area increments (Campo and Vázquez Yanes, 2004). However, after three years of

fertilization in a Costa Rican forest, wood production was unchanged following either N or P fertilization (Waring et al., 2019). Instead, Waring et al. (2019) found that fine root production increased in response to P fertilization, but not following N fertilization.

Increases in fine root production in response to P fertilization are surprising in light of resource limitation theory (Bloom et al., 1985; Chapin et al., 1987). This theory stipulates that trees should grow the tissue type (leaves, wood, fine roots) that would

increase uptake rates of the most limiting resource to achieve optimal partitioning. For example, plants should allocate

relatively more biomass to leaves when light is limiting, and relatively more to roots when nutrients or water are limiting. According to this theory, if fine root biomass is the limiting factor for nutrient acquisition, then we would expect fine root production to decrease as soil nutrients increased, and vice-versa. Some temperate forests have responded in this way to N fertilization (George & Seith, 1998; Wang et al., 2017; Lugli et al., 2021) and this response has also been observed in

herbaceous species (Shipley and Meziane 2002). But these ideas do not fully account for construction costs of new tissues. For example, allocation to fine roots may not be favored if the potential gains in nutrient uptake resulting from increased fine root biomass are less than the nutrient cost of constructing that biomass. Such a situation would be more likely to occur if soil nutrient supply is relatively low.

Models that simulate the terrestrial carbon sink must correctly simulate the effects of P availability on biomass allocation.

Otherwise, if simulated allocation were biased, the simulated carbon sink would likely also be biased because wood residence time is much longer than that of leaves or fine roots. Moreover, as most soil carbon is derived from roots rather than aboveground tissues (Jackson et al., 2017), the soil carbon pool is also likely sensitive to plant biomass allocation. Finally, incorrect allocation would also likely lead to biases in simulations of ecosystem functioning. Despite the number of P-enabled models now exist (e.g. CASACNP, Wang et al., 2010; JSBACH, Goll et al., 2012; CLM-CNP, Yang et al., 2014; ORCHIDEE-

CNP, Goll et al., 2017; QUINCY, Thum et al., 2019; ED2, Medvigy et al., 2019; JULES-CNP, Nakhavali et al., 2022), the simulated effects of P availability on relative allocation to leaves, wood and fine roots have rarely been analyzed.

A further difficulty is that models of the carbon sink typically simulate decades to centuries, but most forest nutrient fertilization experiments have only been carried out for a few years (Wright 2019). A longer-term experimental perspective could be instructive, especially in secondary forests. In these forests, nutrient demand can change rapidly over the course of a

few decades (Batterman et al., 2013; Waring et al., 2015), and changing nutrient demand may lead to changes in allocation strategies. While models have rarely be validated on these time scales, models can nevertheless be used for sensitivity analysis. The results from sensitivity analyses can then be used to pinpoint potentially important processes and to suggest hypotheses for future field experiments.

The objective of this study was to use both a model and an experiment to better understand how relative allocation varies with

nutrient availability. Our model was the ED2 vegetation demographic model that now includes N and P cycling (Medvigy et al., 2019). The experiment involved N and P fertilization in a secondary tropical dry forest in Guanacaste, Costa Rica at Estación Experimental Forestal Horizontes (https://www.acguanacaste.ac.cr) (Waring et al., 2019). We implemented a new allocation scheme in which root production was made dependent on soil P concentration. We carried out model validation and hypothesized that biomass production would be best simulated under the assumption that relative allocation to fine roots is

positively correlated with soil P. We also carried out a sensitivity analysis to determine how allocation parameterization affected simulations on time scales ranging from three to thirty years.

## 2 Materials and Methods





## 2.1. Field Site and Observations

A nutrient fertilization experiment has been ongoing since 2015 at Estación Experimental Forestal Horizontes (10.712N, 85.594W) in Guanacaste, Costa Rica. The experimental design is fully described in Waring et al. (2019) and is summarized here. The site is embedded within an approximately 30-year-old regenerating tropical dry forest, where mean annual temperature is about 25°C and mean annual precipitation is about 1700mm. Precipitation has strong seasonality with most of rain falling between May and November, and has high interannual variability typically associated with El Niño Southern Oscillation. Soils are mainly Andic and Typic Haplustepts (Alfaro et al., 2001), with high percentage of clay (38±1%) and a total N:P of 8.3±0.4. The majority of trees in Horizontes are deciduous and arbuscular mycorrhizal (Hayward and Horton, 2014), and the distribution of plant functional groups are analogous to nearby regenerating forests (Powers and Tiffin, 2010). Although it is a secondary forest, this region has notable biodiversity (60 tree species from 23 families within a 1-ha area in the experimental plots), including many nitrogen-fixing legumes (average of 17±4% stand basal area, and range of 1-53%).

The experiment consists of 16 25m×25m plots, each containing approximately 70 stems ≥5 cm diameter at breast height (DBH). Plots were randomly assigned to one of four treatments: control, nitrogen addition (150 kg N ha$^{-1}$ yr$^{-1}$, urea solution), phosphorus addition (45 kg P ha$^{-1}$ yr$^{-1}$, phosphoric acid solution), or addition of N and P together. Nutrient addition started in June 2015, and was carried out by spraying the solutions three times per year (early, middle and late wet season). Leaf production was measured using litter traps, wood production was measured using tree diameter measurements and allometric equations and fine root production was measured using root in-growth cores. Other measurements included soil $NO_3$ and $NH_4$, soil $PO_4$, and tree mortality.

Results from this study, covering the years 2015-2017, have been reported in Waring et al. (2019). In brief, leaf production did not vary by treatment or by year. Wood production varied by year but not by treatment. Fine root production varied by both treatment and year, with fine root production being about 40% larger in the +P and +NP treatments than in the control or +N treatment.

## 2.2 Model Description

Our model simulations were conducted using the ED2 model (Medvigy et al. 2009, 2019; Longo et al., 2019a). ED2 is a vegetation demographic model that simulates the dynamics of plant cohorts (Fisher et al., 2018). The model has recently been validated in both tropical dry forests (Xu et al., 2016; Medvigy et al., 2019; Schwartz et al., 2022) and tropical moist forests (Levy-Varon et al., 2019; Longo et al., 2019b; Xu et al., 2021). The source code is publicly available on GitHub (https://github.com/EDmodel/ED2).

Each cohort is specified by its plant functional type (PFT), physical dimensions (height and DBH), and stem number density. Each cohort's PFT designation is constant, but physical dimensions and stem number density vary over time. The three critical





demographic processes simulated by the model are growth (increases in physical dimensions), mortality (decreases in stem density), and recruitment (creation of new cohorts). Cohort biomass compartments include leaf, wood, fine root, and non-structural biomass. New photosynthate gets added to the non-structural pool, and respiratory costs are also debited from this pool. Growth occurs when C, N, and P move from their respective non-structural pools to the leaf, wood, and fine root pools. Wood biomass and maximum leaf biomass are related via an allometric relationship (Longo et al., 2019a). In previously published versions of the model, maximum fine root biomass was assumed to be directly proportional to maximum leaf biomass (Longo et al., 2019a), but here we explore various alternatives, as described below. Phenological variation leads to sub-maximal leaf and fine root biomass (Xu et al., 2016).

Simulated growth can be constrained by nutrients (Medvigy et al., 2019). Structural tissues (leaf, wood and fine root) have fixed C:N:P stoichiometries; however, the non-structural pools do not have a fixed stoichiometry. When C, N, and P are initially acquired, they accumulate in their respective non-structural pool. Allocation to leaves and fine roots is done simultaneously on a daily time step. This allocation step can either be limited by the supply of any of the nonstructural pools (C, N, or P). It can also be limited by the maximum leaf and fine root biomass as determined by allometric equations. Whatever remains in the nonstructural pools at the end of each month is used to simultaneously generate new wood and reproductive tissues; this process is only limited by the sizes of the nonstructural pools (Medvigy et al., 2019; Longo et al., 2019a). Some PFTs are capable of symbiotic N fixation (Levy-Varon et al., 2019; Medvigy et al., 2019). The model's approach to soil biogeochemistry explicitly includes microbial mechanisms of soil organic matter decomposition (i.e., enzymatic catalysis) (Wang et al., 2013). Nutrient competition between plant and microbes (for N) and between plants, microbes and mineral surfaces (for P) is calculated using an equilibrium chemistry approximation (Zhu et al., 2016). Growth can also be constrained by water (Xu et al., 2016). As leaf water potentials become increasingly negative, photosynthesis and stomatal conductance are down-regulated. Drought deciduousness is triggered when leaf water potentials are persistently below the turgor loss point (Xu et al., 2016).

The model implements mortality by reducing cohort stem density (Longo et al., 2019a). Each PFT has a baseline mortality rate that is applied to all corresponding cohorts. In addition, cohort-level mortality rates increase rapidly if respiration persistently exceeds photosynthesis. Finally, recruitment consists of the creation of a new cohort at minimum height (typically set to 1-2 m). Recruitment is driven both by external seed rain and the reproduction investment of local cohorts.

## 2.3 Model Modifications

We defined a parameter, *r2l*, which specified the target ratio of fine root biomass to leaf biomass. In previously published versions of the model, *r2l* is a constant. Because several fertilization studies found that fine root production increases with soil P (Waring et al., 2019; Lugli et al., 2021; Cunha et al., 2022), we modified the code so that *r2l* would be related to soil soluble P (*psol*, unit: gP / kg soil) following:

$$r2l = a + b * psol \qquad\qquad (1)$$



where $b$ could be positive, negative or zero. In the remainder of the manuscript, we refer to $b > 0$ parameterizations as "pos" parameterizations, $b < 0$ parameterizations as "neg" parameterizations, and $b = 0$ parameterizations as "const" parameterizations. Our initial model parameterization was a "const" parameterization with $a = 0.3$ (unit: (kgC fine root) / (kgC leaf)) and $b = 0$ (unit: (kgC fine root) / (kgC leaf) * (kg soil) / (gP)), consistent with previous ED2 simulations of tropical dry forests.

**2.4 Simulations**

The purposes of our simulations were model validation and sensitivity analysis (Table 1). Validation required that we focus on the three years (2015-2017) of previously published observations (Waring et al., 2019). We validated the baseline (Medvigy et al., 2019) model parameterization, as well as alternative parameterizations. We carried out sensitivity analysis on both three-year and thirty-year timescales. The three-year timescale was chosen to correspond with the field experiment. The thirty-year

timescale was chosen to see how model sensitivity varied over the course of forest development. A thirty-year simulation would approximately double the age of the forests and would be one order of magnitude longer than the existing experiment.

**Table 1. Description and rationale of model simulations**

| Simulation set | Number of simulations | Allocation schemes | Nutrient input | Analysis period | Rationale |
|---|---|---|---|---|---|
| Baseline | 16, corresponding to 16 plots | const2 ($a$=0.3, $b$=0) | Factorial design (4 control, 4 +N, 4 +P, 4 +NP) | 2015-2017 | Validate the baseline model |
| Alternative parameterizations, short-term | 16 plots * 12 parameterizations | neg1, neg2, neg3 pos1, pos2, pos3 const1, const3, const4, const5, const6, const7 | Factorial design (4 control, 4 +N, 4 +P, 4 +NP) | 2015-2017 | (1) Determine short-term sensitivity of model to parameterization; (2) validate alternative parameterizations |
| Alternative parameterizations, long-term | 16 plots * 7 parameterizations | neg1, neg2, neg3, const2, pos1, pos2, pos3; | Factorial design (4 control, 4 +N, 4 +P, 4 +NP) | 30 years | Determine longer-term sensitivity of model to parameterization |





### 2.4.1 Baseline simulations and validation

We simulated each of the 16 experimental plots using the model's baseline parameterization. The vegetation cohorts in the
model were initialized with in situ measurement of DBH and height data for each individual tree. Soil properties of each site
were initialized with in situ soil state observations following the procedure of Medvigy et al. (2019). Nutrient additions rates
were set to be equivalent to the amount added to each site in the experiment: control, N addition (126 kgN ha$^{-1}$ yr$^{-1}$), P addition
(50 kgP ha$^{-1}$ yr$^{-1}$), or addition of N and P together (Waring et al. 2019). We also applied natural deposition rate of 0.13 kgN
ha$^{-1}$ yr$^{-1}$ and 0.019 kgP ha$^{-1}$ yr$^{-1}$ in all 16 plots. All the simulations were driven by meteorological variables from the ERA5-
Land hourly reanalysis datasets (Muñoz Sabater, 2019). Simulations ran from January 2013 until April 2018, and we analyzed
the same time period as the field measurements, 2015-2017. The first two years were discarded as spin-up.

Simulations and observations were compared both qualitatively and quantitatively. For qualitative validation, we emphasized
(1) production averaged over treatments and/or years and (2) variation in production across treatments. For quantitative
validation, we used Student's t-tests to assess whether the simulations and the observations had the same means. Linear
regression was used to assess whether observed variation in production across treatments and years was accurately simulated.
In all statistical tests, we applied $p < 0.05$ as the threshold for statistical significance.

We were only able to use Student's t-tests to evaluate leaf and wood production. The reason is that $t$-tests require multiple
replicates that are all assumed to be drawn from the same normal distribution. For leaf production, we had 12 replicates (three
years times four treatments). For wood production, previous analysis had shown that data from different years had statistically
significant variation; i.e., they cannot be assumed to be drawn from the same distribution (Waring et al., 2019). We therefore
averaged wood production over the three years, leaving us with four replicates. These four replicates correspond to the four
treatments and they are suitable for our t-test because previous analysis did not identify a statistically significant treatment
effect (Waring et al., 2019). Prior to applying t-tests to leaf or wood production, we confirmed normality with the Shapiro-
Wilk test ($p < 0.05$). We also assessed equality of variances with Welch's test.

For fine root production, previous analysis demonstrated significant year and treatment effects (Waring et al., 2019). Thus, the
different fine root production observations need to be assumed to have come from distributions, and a $t$-test would be
inappropriate. Instead, linear regression was used to assess the model's ability to capture variation across treatments and years.
We considered a parameterization to be "validated" if the regression intercept was consistent with zero and the slope was
consistent with one. In principle, we could also use linear regression to validate our simulations of wood production across
years. However, with only three years of data, the statistical power of this test would be very weak and we did not perform it.
Neither did we perform this test in our evaluation of leaf production because no variation across treatments or years was
observed.





### 2.4.2 Alternative parameterizations

We analyzed the sensitivity of production (leaf, fine root, wood, total) to allocation parameterization. First, we linked variation
in $a$ to variation in $b$ such that all parameter combinations would yield approximately the same $r2l$ for the control (unfertilized)
plots. Preliminary simulations suggested that the following constraint would be appropriate:

$$0.3 = a + \frac{b}{200} \tag{2}$$

Under this constraint, different $a$-$b$ combinations would yield about the same $r2l$ for the control plots, but different $r2l$ for the
P fertilized plots (larger $r2l$ in the fertilized plots for "pos" parametrizations and smaller for "neg" parametrizations). Larger
absolute values of $b$ indicate greater sensitivity of allocation to $psol$. The largest value of $b$ that we tested was 60 (kgC fine
root) / (kgC leaf) * (kg soil) / (gP), which corresponded to $a=0$ and also set $r2l=0$ when $psol=0$ (Table 2). We regarded this
parameter setting as an end member case. For complementarity, the lowest value of $b$ that we tested was -60 (kgC fine root) /
(kgC leaf) * (kg soil) / (gP). Altogether, we tested a total of seven $a$-$b$ pairs (Table 2), the same number of values used in the
sensitivity tests of LeBauer et al. (2013).

We also tested sensitivity to a constant $r2l$ that did not respond to $psol$. That is, we varied $a$ while keeping $b$ fixed at zero. The
particular values that we tested ranged from $a=0.2$ to 0.8 (kgC fine root) / (kgC leaf) (Table 2).

### 2.4.3 Validation of sensitivity analysis of alternative parameterizations

The alternative parameterizations were validated in the same way as the baseline simulations.

Sensitivity analysis was carried on three-year and thirty-year timescales. Seven parameter settings were chosen, and response
variables (leaf, wood, fine root production) were plotted against parameter value. Cubic splines were fit through these points,
and we also determined coefficients of variation. For the thirty-year simulations, we had to prescribe meteorological forcing
and fertilization rates for years beyond 2018. For these years, the meteorological forcing was obtained by recycling the 2009-
2018 observations. N and P fertilization rates were maintained as they were in all other simulations; thus, these simulations
constituted virtual 30-year fertilization experiments.

**Table 2. Parameter values for each allocation scheme**

| Parameterization | Value of $a$ (unit: (kgC fine root) / (kgC leaf)) | Value of $b$ (unit: (kgC fine root) / (kgC leaf) * (kg soil) / (gP)) |
|---|---|---|
| neg1 | 0.6 | – 60 |



| neg2 | 0.5 | – 40 |
| neg3 | 0.4 | – 20 |
| const2 | 0.3 | 0 |
| pos1 | 0.2 | 20 |
| pos2 | 0.1 | 40 |
| pos3 | 0 | 60 |
| const1 | 0.2 | 0 |
| const3 | 0.4 | 0 |
| const4 | 0.5 | 0 |
| const5 | 0.6 | 0 |
| const6 | 0.7 | 0 |
| const7 | 0.8 | 0 |

## 3 Results

### 3.1 Baseline simulation validation

We first qualitatively evaluate the baseline model simulation ($a$=0.3, $b$=0). Variation across treatments and years is shown in Fig. 1. The magnitude of leaf production was similar in simulations and observations, though the simulations had a larger

range of values (Fig. 1A). The simulations and observations agreed that wood production was smallest in the strong ENSO year of 2015, but disagreed as to whether maximum production occurred in 2016 or 2017 (Fig. 1B). Overall, the model somewhat overestimated wood production. Fine root production had a much larger bias than leaf or wood production, especially in 2015-2016 (Fig. 1C). The baseline simulations did not appear to capture the observed treatment effect (higher fine root production in +P and +NP than in the control and +N plots).

We also compared other simulated values to observations. The simulated stem mortality was close to observations in each of three years, including 2015, when stem mortality was relatively large (Table 3). Over the three years, simulated stem mortality





in +NP plots was about 1.5 times larger than the other treatments, consistent with observations. Simulated and (temporally sparse) observations of plant available nutrients are shown in Fig. 2. The observed soluble P, $NH_4$, and $NO_3$ fell within the range of what was simulated. Both the simulations and the observations show a strong effect of P fertilization on soluble P. In

the simulations, most of the peaks associated with the pulse inputs of P are clearly visible. By contrast, similar peaks are much less apparent in $NH_4$ and $NO_3$, and fertilization had a weak impact on their concentrations.





**Figure 1. Comparison of simulated and observed (a) leaf, (b) wood and (c) fine root production for each year-treatment combination.**

**Table 3. Comparison of simulated and observed annual stem mortality. The notation "+NP/others" indicates the ratio of the result from the +NP treatments to the average result from the control, +N, and +P treatments.**

| Stem mortality | 2015 | 2016 | 2017 | +NP/others |
|---|---|---|---|---|
| obs | 10.6% | 6.0% | 4.6% | 1.3-1.8 |
| neg1 | 13.5% | 6.3% | 4.8% | 0.9 |
| neg2 | 12.2% | 6.1% | 4.8% | 1.0 |
| neg3 | 10.5% | 5.8% | 4.8% | 1.3 |
| const2 | 10.7% | 6.3% | 4.7% | 1.5 |
| pos1 | 9.1% | 5.7% | 4.7% | 1.4 |
| pos2 | 10.3% | 5.8% | 4.8% | 1.4 |
| pos3 | 11.8% | 6.4% | 4.7% | 1.2 |
| const1 | 9.3% | 6.0% | 4.9% | 1.4 |
| const3 | 11.6% | 5.9% | 4.9% | 1.2 |
| const4 | 13.5% | 5.9% | 4.8% | 1.1 |
| const5 | 15.4% | 5.9% | 4.8% | 1.0 |
| const6 | 17.4% | 6.3% | 4.7% | 0.9 |
| const7 | 19.8% | 6.5% | 4.6% | 0.9 |





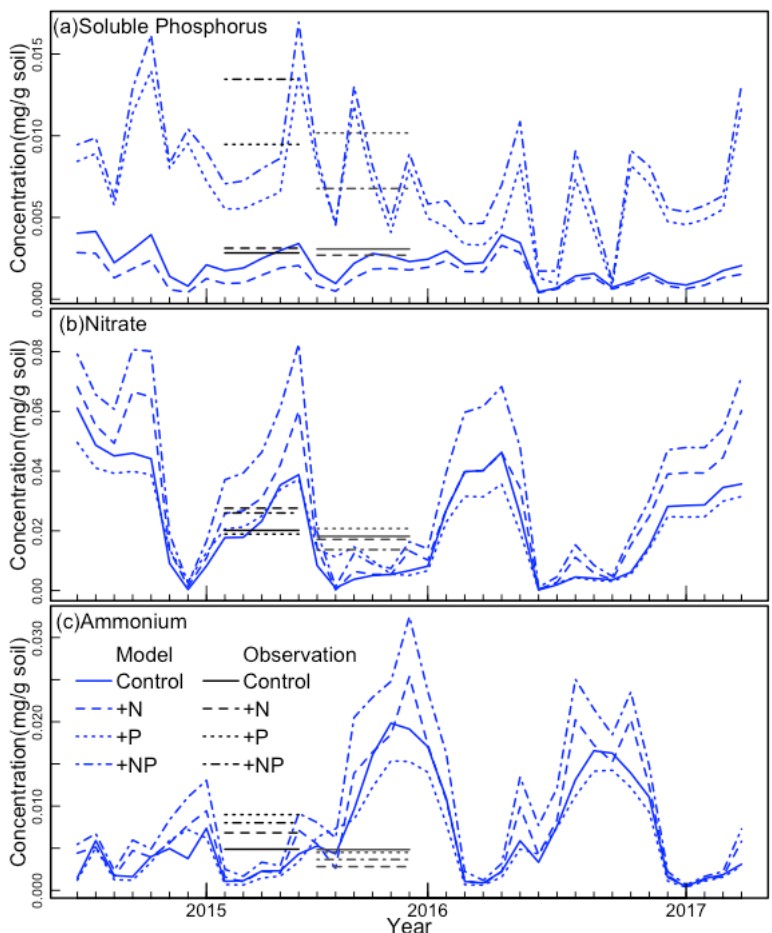

**Figure 2. The continuous curves show the model-simulated (a) soil soluble phosphorus, (b) nitrate, and (c) ammonium concentrations. The black horizontal line segments indicate the observed averages from the wet and dry seasons in 2016; observations were not available for other periods.**

**3.2 Alternative Parameterizations: Three Year Sensitivity analysis**

We then analysed the sensitivity of the simulations to parameterization (Fig. 3). Different treatments were analyzed separately. In the control and +N treatments, leaf production was relatively insensitive to parameterization, wood production increased from the "neg" to the "pos" parameterizations, and fine root production decreased from the "neg" to the "pos" parameterizations. The sensitivity of total production was similar to that of wood production. By contrast, in the +P and +NP

treatments, the different components of production did not vary monotonically. Rather, leaf and fine root production were minimized and wood production was maximized with the "const" scheme.

Fine root production had a larger CV than leaf and wood production, and also showed substantial variation across treatments. Wood production had a larger CV than leaf production, but the leaf production CV was more sensitive to treatment.





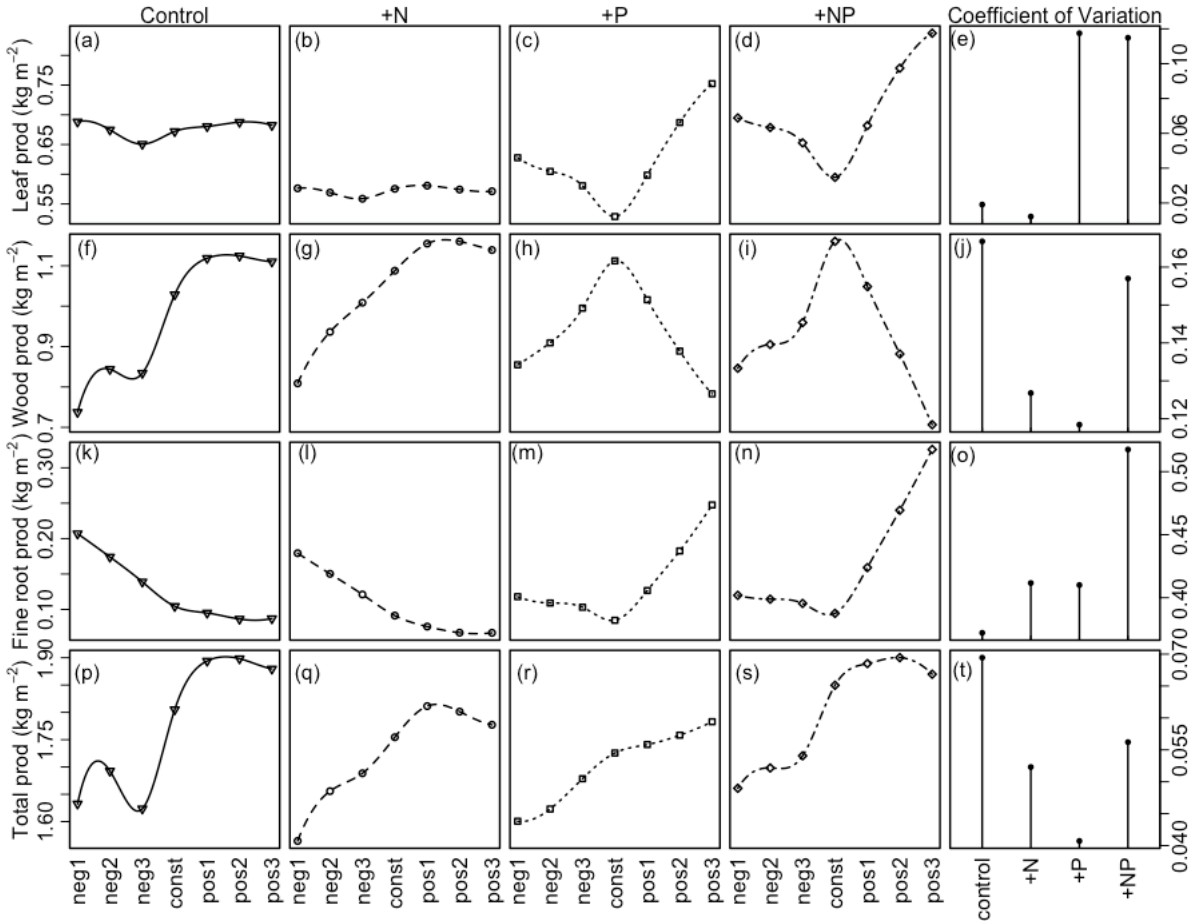

**Figure 3. Variation of simulated leaf, wood, fine root and total biomass production among treatments under different parameterizations. The lines in left four columns represent fitted cubic spline. Figure (e), (j), (o), (t) represent the coefficient of variation.**

### 3.3 Alternative Parameterizations: Model Validation

Our "neg" and "pos" parameterizations all slightly overestimated the observed leaf production, while the "const" parameterizations underestimated the observed leaf production (Fig. 4a). This result is consistent with our sensitivity analysis, which showed that the "const" parameterization minimized leaf production. Despite these differences, the observed and simulated means were significantly different ($p < 0.05$) in only two cases, pos3 and const6 (Table 4). All simulations showed greater variability than the observations (Fig. 4a). Wood production was generally overestimated by the model (Fig. 4b). For the simulations with $b=0$ and relatively small values of $a$, the model and observations differed significantly (Table 4). Simulations with larger negative or positive values of $b$ were consistent with observations (Table 4). All "neg" and "pos" parameterizations underestimated fine root production, while "const" parameterizations varied in the sign of their bias (Fig.





4c). We did not compare simulated and observed mean fine root production using t-tests because the different observations are known to be drawn from different distributions (see Methods).

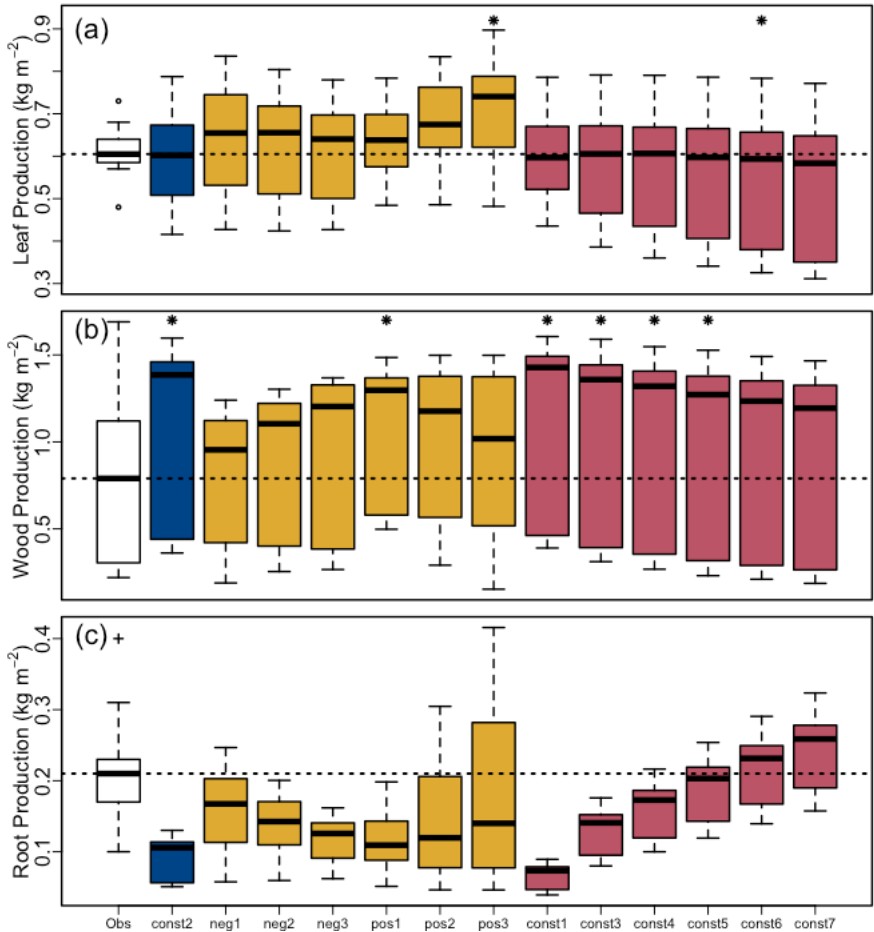

**Figure 4. (a) Leaf production for the observations and different model parameterizations. The horizontal dashed line represents the median of observations. Panels (b) and (c) are similar, but show wood production and fine root production, respectively. A "*" indicates that a simulation is significantly different from observations. In the case of fine root production, the "+" indicates no statistical test was performed.**

**Table 4. P-values of the tests for difference in means between simulated and observed leaf and wood production. Significantly**
**different means are indicated with *. This statistical test was not applied to fine root production.**

| Parameterization | $a$ | $b$ | Leaf production: t-test | Wood production: t-test |
|---|---|---|---|---|
| neg1 | 0.6 | -60 | 0.403 | 0.642 |
| neg2 | 0.5 | -40 | 0.616 | 0.223 |
| neg3 | 0.4 | -20 | 0.997 | 0.102 |
| const2 | 0.3 | 0 | 0.628 | 0.010* |
| pos1 | 0.2 | 20 | 0.451 | 0.013* |





| pos2 | 0.1 | 40 | 0.053 | 0.073 |
| pos3 | 0 | 60 | 0.022* | 0.279 |
| const1 | 0.2 | 0 | 0.705 | 0.007* |
| const3 | 0.4 | 0 | 0.469 | 0.015* |
| const4 | 0.5 | 0 | 0.359 | 0.025* |
| const5 | 0.6 | 0 | 0.265 | 0.043* |
| const6 | 0.7 | 0 | 0.034* | 0.216 |
| const7 | 0.8 | 0 | 0.131 | 0.106 |

Because fine root production showed both year and treatment effects, we regressed simulated fine root production on the observations. Among all the parameterizations, only pos2 and pos3 had a slope consistent with one and intercept consistent with zero (Fig. 5, Table 5). Simulation pos1 also had an intercept consistent with zero, but it had a slope that was significantly

less than one. Other simulations performed more poorly (Table 5). In terms of mortality, the largest biases were found in simulations const5, const6 and const7, which had relatively large values of $a$ combined with $b$=0 (Table 3).

**Table 5. Statistics from the fine root regressions.**

| Parameterization | $a$ | $b$ | Intercept estimate | Intercept standard error | p-value | Slope estimate | Slope standard error | p-value |
|---|---|---|---|---|---|---|---|---|
| neg1 | 0.6 | -60 | 0.25 | 0.06 | 0.002 | -0.44 | 0.28 | 0.145 |
| neg2 | 0.5 | -40 | 0.20 | 0.05 | 0.001 | -0.32 | 0.22 | 0.170 |
| neg3 | 0.4 | -20 | 0.15 | 0.04 | 0.002 | -0.15 | 0.17 | 0.374 |
| const2 | 0.3 | 0 | 0.10 | 0.03 | 0.011 | -0.04 | 0.15 | 0.782 |
| pos1 | 0.2 | 20 | 0.04 | 0.04 | 0.358 | 0.37 | 0.19 | 0.075 |
| pos2 | 0.1 | 40 | -0.02 | 0.07 | 0.755 | 0.82 | 0.34 | 0.039 |
| pos3 | 0 | 60 | -0.07 | 0.11 | 0.553 | 1.23 | 0.53 | 0.042 |
| const1 | 0.2 | 0 | 0.07 | 0.02 | 0.005 | -0.02 | 0.09 | 0.817 |
| const3 | 0.4 | 0 | 0.14 | 0.04 | 0.003 | -0.03 | 0.17 | 0.841 |
| const4 | 0.5 | 0 | 0.17 | 0.04 | 0.003 | -0.04 | 0.2 | 0.863 |
| const5 | 0.6 | 0 | 0.20 | 0.05 | 0.002 | -0.03 | 0.23 | 0.894 |
| const6 | 0.7 | 0 | 0.22 | 0.05 | 0.002 | -0.02 | 0.26 | 0.936 |
| const7 | 0.8 | 0 | 0.24 | 0.06 | 0.002 | -0.003 | 0.27 | 0.991 |



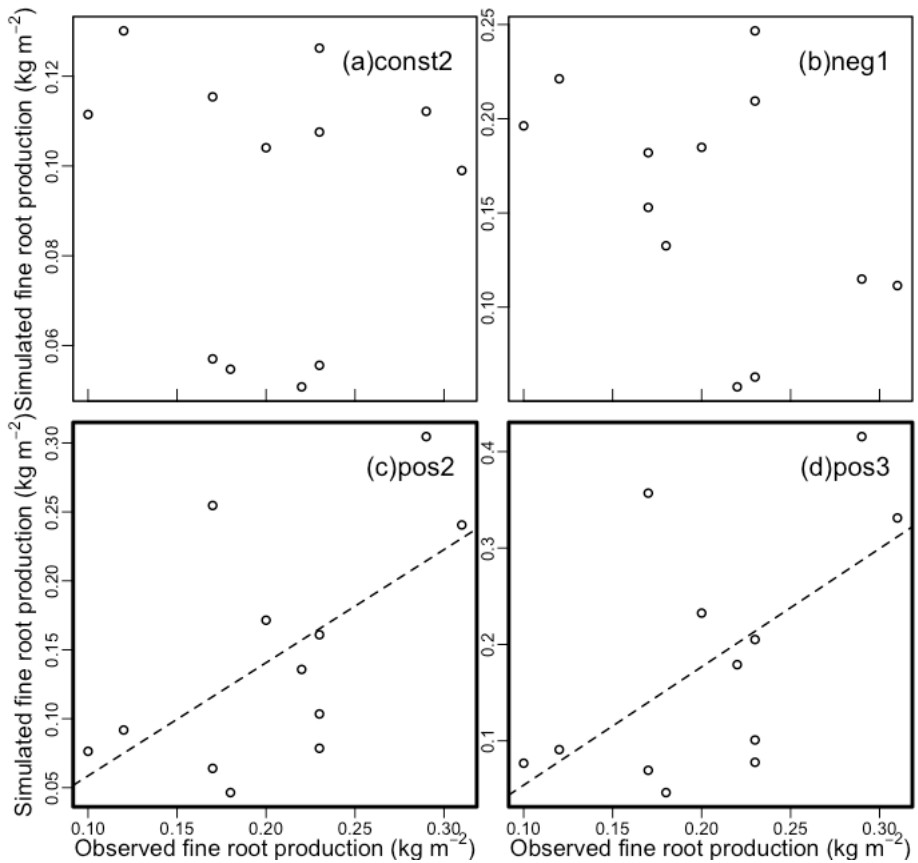

**Figure 5. Comparison of simulated and observed fine production for 4 representative parameterizations. (a)const2 and (b)neg1 show**
**typical patterns of "const" and "neg" parameterizations. Only (c)pos2 and (d)pos3 have p-values less than 0.05.**

### 3.4 Alternative Parameterizations: Thirty-Year Sensitivity analysis

Allocation parameters had a large impact on simulated AGB accumulation, as seen in our multi-decadal simulations (Fig. 6a,
e). Over a 30-year timeframe, the baseline parameterization consistently simulated the more biomass throughout the study
period than the "neg" and "pos" parameterizations. This pattern of simulated AGB resulted from the patterns of AGB growth
and mortality, as the "neg" and "pos" parametrizations typically led to lower AGB growth (Fig. 6b, f) and higher AGB
mortality (Fig. 6c, g) than the baseline parameterization. However, the relative ranking of the "neg" and "pos" parametrizations
varied throughout the 30 years. During approximately the first decade, the "neg" parametrizations had the least plant biomass.
For the rest of the time period, biomass in the pos3 simulation declined markedly, and this simulation ended up with the least
amount of biomass. Also notable in all simulations were the large biomass dips that occurred in approximately years 10 and
20. These biomass dips coincided with the driest year in meteorological forcing.

The relationship between biomass growth and P limitation was complex. In our model, we assessed P limitation by considering
the non-structural P pool. For each cohort, this pool increased when the cohort took P out of the soil and it decreased when P





was deployed to build structural tissue. The pool has a maximum size, dependent on cohort DBH. If the ratio of actual non-structural P to maximum non-structural P were equal to one, then the cohort would be able to rapidly make up for any P used

for growth and there would be no P limitation. Values less than one would indicate P limitation, and a value of zero would represent an extreme case in which a cohort has no P to support growth. There was a clear distinction across simulations, with the "pos" parametrizations having the lowest values (strongest P limitation) and the baseline and "neg" parameterizations having values closer to one (Fig. 6d, h). For all simulations, this ratio exhibited substantial interannual variability, but no long-term trend.

**Figure 6. Time series of simulated (a)AGB, (b)AGB growth, and (c)AGB mortality over 30 years under different parameterizations. The ratio of nonstructural P to maximum nonstructural P ("Pstorage/PstorageMax"), an index of P limitation, is shown in panel (d). Also shown are (e)AGB, (f)AGB growth, (g)AGB mortality, and (h) ratio of nonstructural P to maximum nonstructural P at the end of the thirty-year simulation plotted against parameterization. Cubic splines connect the points in (e)-(h).**

To better understand the results in Fig. 6, and to evaluate the P fertilization effect, we analyzed the P-fertilized plots (+P and +NP) separately from the others (control and +N). In the control and +N treatments, the positive correlation parameterizations had larger AGB than negative correlation parameterizations throughout most of the simulation period (Fig. 7a), and well





relatively large AGB growth (Fig. 7b) and intermediate AGB mortality (Fig. 7c). The large AGB growth rates in positive correlation parameterizations were not the result of there being little P limitation in that simulation (Fig. 7d). In fact, positive
correlation parameterizations tended to have more P limitation than negative correlation parameterizations. To explain why positive correlation parameterizations had larger AGB growth despite having more P limitation, we explored other aspects of the simulations. In particular, relative allocation to fine roots affected water capture in that the "neg" parameterizations had less transpiration than the others (Fig. 8).

The +P and +NP treatments showed much more sensitivity of AGB to parameterization (Fig. 7e) than the control and +N
treatment (Fig. 7a). This difference occurred because P fertilization reduced the AGB growth (Fig. 7f) and increased mortality (Fig. 7g) in the "pos" schemes relative to the other schemes. This result occurred despite there being relatively little P limitation in the P-fertilized plots (Fig. 7h).

**Figure 7. Effect of P fertilization on AGB (a,e), AGB growth (b,f), AGB mortality (c,g) and the ratio of nonstructural P to maximum**
**nonstructural P (Pstorage/PstorageMax) (d,h) during 30-year simulations. Panels (a)-(d) show the average over the control and +N treatments and panels (e)-(h) shown the average of the +P and +NP treatments.**



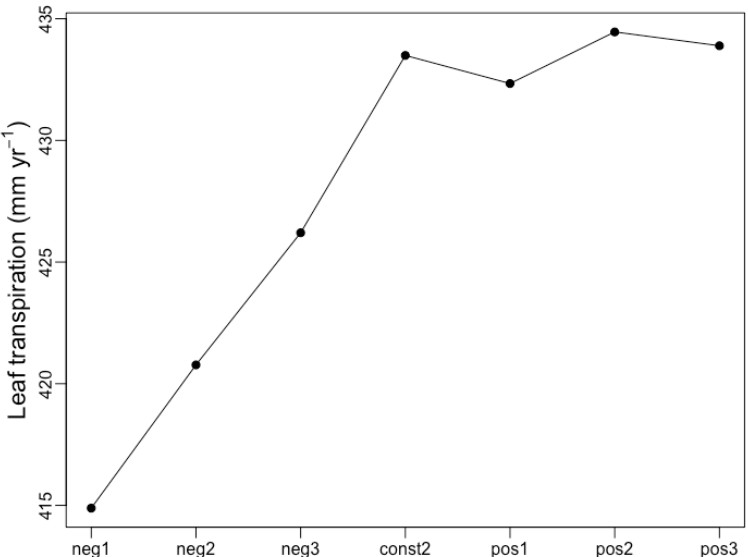

**Figure 8. Average leaf transpiration of 8 control and +N plots over 30-year simulations. Only the wet months (May-November) in each year were included in the analysis.**

**4 Discussion**

Soil nutrients can regulate plant biomass production in terrestrial ecosystems (LeBauer and Treseder, 2008; Hou et al., 2020). The control that nutrients exert on carbon partitioning among different types of plant tissues is drawing increasing attention because it can strongly affect long-term ecosystem carbon accumulation and loss (Gessler et al., 2016). This study focused on the effect of soil soluble P on carbon partitioning. We evaluated different parameterizations within the ED2 model, and

compared model results to observed carbon partitioning at a fertilization experiment site in Costa Rican tropical dry forest (Waring et al., 2019). The results presented here demonstrate the importance of a nutrient-sensitive allocation parameterization. In particular, we found that the model simulated the most realistic overall partitioning of biomass production when relative allocation to fine roots was positively correlated with soluble soil P (a "pos" parameterization). Analysis of multi-decadal simulations suggests that under certain circumstances these "pos" parametrizations can lead to greater aboveground carbon

accumulation than parameterizations where relative allocation to fine roots was independent of soil P ("const" parameterizations) or negatively correlated with soil P ("neg" parameterizations).

**4.1 Model validation**

ED2 has long included N dynamics as an option (Moorcroft et al., 2001), but published simulations rarely had this option activated. Recently, Medvigy et al. (2019) introduced a new representation of N and P dynamics into ED2, based on microbial

model of Wang et al. (2013) and the nutrient competition model of Zhu et al. (2016). In the Medvigy et al. (2019) parameterization, relative allocation to fine roots was unaffected by nutrients (i.e., the model employed a "const"



parameterization). Here, we found that this version of the model simulated reasonable leaf and wood biomass production, especially when averaged over a three-year time frame (Figs. 1, 2). These successful predictions of leaf and wood production are consistent with previous ED2 simulations of tropical forests (Xu et al., 2016, 2021; Levy-Varon et al., 2019; Longo et al.,

2019a; Longo et al., 2019b). An important feature of our analysis is that we additionally validated the model's simulation of fine root production. We found that the baseline parameterization resulted in an underestimate of fine root production and that "const" parameterizations in general could not simulate the observed stimulation of fine root production by P fertilization (Fig. 1C; Table 5). At this point, we cannot say whether ED2 would generate similar biases in fine root production at other tropical forest sites. More observations of biomass partitioning (including fine root production) under P fertilization would be helpful

for testing the model, and such observations are becoming increasingly available (Yuan and Chen, 2012; Wright, 2019; Cunha et al., 2022).

Besides our baseline parameterization, we evaluated other "const" parameterizations as well as "pos" and "neg" parametrizations. Most parameterizations simulated leaf production consistent with observations and about half simulated wood production consistent with observations (Fig. 4, Table 4). However, most parameterizations failed to simulate root

production that was consistent with the observations (Fig. 5, Table 5). As a result, only one of the 13 parameterizations that we tested was able to simultaneously simulate leaf, wood and fine production consistent with the observations. This particular parameterization was a "pos" parametrization with a moderate sensitivity of fine root relative allocation to soil soluble P. Importantly, it was also reasonably accurate in simulating tree mortality and soil nutrient concentrations (Table 3). It will be interesting to see if this "pos" parametrization would be consistent with longer-term observations of production, biomass

partitioning and mortality. We will soon be able to test that point, as the fertilization experiment first reported by Waring et al. (2019) is ongoing and the data are currently being analyzed. It will also be important to test "pos" parametrizations at other tropical forest sites. For example, Lugli et al. (2020) found increasing fine root production with increasing soil P but did not report relative allocation of fine roots to leaves, so we do not know if their relationship was due to an increase in total production (including fine root production) or whether biomass partitioning changed. Cunha et al. (2022) reported strong increase of NPP

exclusively with P fertilization, and fine root production was found to have greater response to P increase than canopy production (29% and 19%, respectively). Their findings showed a preferred biomass allocation to root when P availability increases, which is what our model with a "pos" parametrization simulates.

These findings are surprising because multiple limitation theory would predict that increases in soil P would result in decreased allocation to fine roots. One potential explanation is that soil P supply, not fine root biomass, limited P uptake in the unfertilized

plots. For example, in the complete absence of soil P, P acquisition would be zero regardless of fine root biomass. The optimal amount of fine root biomass (with respect to P acquisition) would be zero in order to avoid construction costs. As soil P increases above zero, the optimal amount of fine root biomass would also increase. An analogy would be "rain roots" that are produced by some species after rain events (Nobel et al., 1990).





Because this study is focused largely on model validation and sensitivity, we took the observational data at face value.
However, the observational data can also have biases that would impact our interpretations. In particular, fine root stock and
loss are difficult to measure accurately in forests (Clark et al., 2001), largely due to highly uncertain spatial and temporal
variability in fine root biomass (Finér et al., 2011) and rooting depth (Paz et al., 2015). In tropical forests, maximum root
length is often longer than the depth of ingrowth cores in Waring et al. (2019) (Canadell et al. 1996), implying that field
measurements underestimate root production. Further, addition of P to the soil surface could have caused roots to proliferate
at the surface, at the expense of deeper roots. Further field experiments are necessary to understand potential changes in root
vertical distributions. Finally, estimates of plant allocation of carbohydrates to mycorrhizae are rare and difficult to obtain, and
were not made by Waring et al. (2019). It is possible that P fertilization led to reduced allocation to mycorrhizae and increases
in both the number and average diameter of roots, while reducing overall belowground allocation. It should be noted that these
biases and our "supply-limited" hypothesis are not mutually exclusive.

### 4.2 Three-Year Sensitivity Analysis

Our approach to sensitivity analysis was to focus on the sensitivity of several output variables (leaf, wood and fine root
production) simultaneously against one input parameter that determined the fine root to leaf ratio. By contrast, other studies
using ED2 have focused on one output variable and multiple inputs (LeBauer et al., 2013; Levy-Varon et al., 2019; Medvigy
et al., 2019). Our sensitivity analysis of leaf, wood, fine root and total production showed distinct responses for the different
production measures (Fig. 3). Of these, fine root production had the largest CV because its average magnitude was smallest.
Our results also varied considerably depending on whether P fertilization was applied, underscoring the importance of
environmental context for sensitivity analysis.

### 4.3 Thirty-Year Sensitivity Analysis

In our 30-year simulations, we found that the effects of fine root allocation parameterization on AGB depended on fertilization
treatment. In plots not fertilized with P, the "neg" simulations had less growth, more mortality and less AGB accumulation
than either the "const" or "pos" simulations (Fig. 7). To understand why, it is important to consider that fine roots facilitate
the acquisition of water as well as P. Different allocation parameterizations led to different amounts of water uptake and thus
transpiration (Fig. 8). Overall, the low AGB accumulation associated with the "neg" simulations provides an explanation for
why the observed fine root production was inconsistent with a "neg" parameterization. By contrast, the "const" and "pos"
simulations had higher transpiration and had higher aboveground biomass accumulation. Interestingly, the increase in
aboveground biomass did not arise from alleviation of P limitation (indeed, P limitation is largest in the "pos" simulations). It
is notable that a long time series is helpful for seeing this effect, as it is most prominent during extremely dry years.

Long-term simulations under P fertilization were qualitatively different (Fig. 7e-h). Given that the "pos" parametrizations were
most consistent with the short-term observations, we were surprised to see that the "pos" parametrizations led to much less

AGB accumulation in the long term then either the "neg" or "const" parameterizations. In the "pos" parametrizations, fertilization drove very high P concentrations and thus very high relative allocation to fine roots. This allocation to fine roots was inevitably costly in terms of root respiration and turnover because the models sets both of these processes to be proportion to fine root biomass. In addition, the marginal benefit of increases in fine allocation are smaller when allocation to fine roots is large than when allocation to fine roots is small. Because P limitation was nearly non-existent under P fertilization, the

benefits of further increasing fine root biomass to scavenge P declined. Thus, the "pos" parametrizations caused the simulations to overcompensate for high P concentrations. At the same time, the "neg" parametrizations also had less AGB accumulation than "const" parameterizations. In the "neg" parametrizations, very high soil P concentrations led to very low allocation to fine roots. This low allocation to fine roots did not much impact P limitation, but it would impact water acquisition. These dynamics also help to explain why "neg" parametrizations were incompatible with the observations of Waring et al. (2019).

**4.4 Towards more sophisticated models**

Going forward, it would be interesting to validate the ability of other models to simulate biomass partitioning at tropical forest fertilization sites. Existing models use a variety of allocation schemes. For example, CLM-CNP (Yang et al., 2014) and JULES-CNP (Nakhavali et al., 2022) use parameterizations similar to our "const" parameterization to control new growth allocation. Other models use schemes that we speculate would function like our "neg" parametrizations. ORCHIDEE-CNP (Goll et al.,

2017) and QUINCY (Thum et al., 2019) applied a pipe theory to partition leaf and root mass, modulated by the most limiting soil available nutrient (and water, for QUINCY). DLEM-CNP applies a method that make allocation co-limited by both N and P (Friedlingstein et al., 1999; Wang et al., 2020). We are not aware of other models using a scheme analogous to our "pos" parametrizations. We suggest that model intercomparison be carried out, especially at tropical nutrient fertilization sites. Our sensitivity analysis also suggests that it will be important to carry out model validation on time scales longer than three years,

as the optimal allocation strategy in the short-term may differ from the optimal allocation strategy in the long-term.

When comparing the influence of different allocation schemes, we made some simplifications to make our analyses more straightforward. For example, we did not account for the effect of N limitation on carbon partitioning. However, such an effect was not observed at our study site (Waring et al., 2019). Neither did we account for the effect of water limitation on carbon partitioning. Carbon partitioning may also depend on community composition (Dybzinski et al., 2011) and it may be temporally

variable (Farrior et al., 2013). We did not account for either of these effects except for the impact of water limitation on tree phenology (Xu et al., 2016). More sophisticated parameterizations that account for these effects should be investigated in future studies.

**5 Conclusion**

The partitioning of the new growth in a forest ecosystem between leaf, wood and fine root pools is a critical aspect ecosystem

functioning and can strongly affect forest carbon budgets (Litton et al., 2007). We applied the nutrient enabled ED2 model in



simulating a fully factorial N and P fertilization experiment conducted in a secondary tropical dry forest in Costa Rica over three years. Some model parameterizations were able to accurately simulate leaf, wood and fine root production, as well as mortality. Surprisingly, these parameterizations all assumed a positive relationship between relative allocation to fine roots and soil P. This result might be expected at relatively low levels of soil P, when increased root growth would lead to larger

construction and maintenance costs but only modest increases in P uptake. Further experimentation is needed to test whether this relationship would hold on to longer time scales and at high P concentrations. Indeed, our sensitivity analysis suggested that other dynamics may be at play on longer time scales. This analysis showed over-allocation to fine roots in long-term, P-fertilized situations. Our findings also suggested the need of more model-data intercomparison, especially with respect to simultaneous measurements of leaf, wood and fine root production. Such analyses will enable us to develop improved model

parameterizations and ultimately better simulations of forest carbon balances.

*Code and data availability.* The most up-to-date source code, post-processing R scripts, and an open discussion forum are available on GitHub at https://github.com/EDmodel/ED2. Field data are available from the Dryad Digital Repository: https://doi.org/10.5061/dryad.mq62g78.

*Author contributions.* SL ran the simulations, carried out the analysis and wrote the first draft of the manuscript. JSP and BGW provided the field data. DM designed the study. All authors contributed to manuscript revisions.

*Competing interests.* DM is a member of the *Biogeosciences* editorial board. The peer-review process was guided by an independent editor, and the authors have no other competing interests to declare.
*Acknowledgements.* DM, BGW, and JSP were supported by the US Department of Energy, Office of Science, Terrestrial Ecosystem Science Program, Award DE-SC0014363. The field plots were maintained by National Science Foundation CAREER Grant DEB-1053237 to JSP.

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
