# Peer review of "Tropical Dry Forest Response to Nutrient Fertilization: A Model Validation and Sensitivity Analysis"

_Biogeosciences, 2022_

## Author Comment (AC1)

Dear Reviewer,

Thank you for the constructive comments. We copied your comments below in blue font; our labeled responses are in ordinary black font. We have labeled our responses for ease in cross-referencing. Responses to your comments are labelled "RC1" and responses to a community reviewer are labeled "CC1". We have not yet received responses from a second reviewer. We welcome further discussion during the discussion period.

General comments

This study integrated a representation of phosphorus-dependent relative allocation to root tissues into the ED2 model. The model was simulated at a site in a tropical dry forest in Costa Rica for +N, +P, and +NP fertilization treatments. Modelled results were compared to empirical observations. The model was then simulated over 30 years to examine the influence of the new process representation over a longer time scale. The comparison between a model and empirical observations of experimental manipulations of nutrient input is very useful for model development. However, it was unclear whether the process that was represented in this study (increasing allocation to fine roots with increasing soil P) is prevalent in systems outside of this site, what its underlying mechanisms are, and how it relates to other central processes (such as the relationships between allocation to fine roots and soil nitrogen or water). Furthermore, the statistics used to establish the results were unclear.

(Response RC1-1): Thank you for your review. We are happy that you found the comparison helpful and are very grateful for your constructive feedback. In our responses below, we have addressed the points of clarification that you raised. We also carried out a new statistical analysis, largely following your advice. We think that incorporation of these changes will result in a greatly improved manuscript.

Specific comments

1. The premise of this study needs to be better established. What is the mechanism underlying increasing allocation to fine roots with increasing soil P?

(Response RC1-2): This study investigated the consequences of three related premises: (a) plants increase fine root production with increasing soil P, (b) plants decrease fine root production with increasing soil P, and (c) plant fine root production is independent of soil P. Regarding (a), several studies have shown that roots proliferate in nutrient-rich patches (Pregitzer et al. 1993; Robinson 1994; Zhang and Forde 1998; Robinson et al. 1999; Fransen et al. 1999; Hodge et al. 1999; Jing et al. 2010; Li et al. 2012). This proliferation may be related to enhanced root morphological plasticity with increasing soil P, which increases nutrient uptake per unit construction cost (Fitter 1994; Eissenstat and Yanai 2002; Zhang et al. 2016). It may also be physiologically adaptive if it allows for faster uptake of nutrients from the soil (Jackson et al. 1990; Hodge et al. 2004). We discussed premise (b) in the original manuscript, lines 58-65. Regarding premise (c), it could be that fine root production depends more on water or N than on P. In this case, relative allocation to fine roots would not be directly sensitive to soil P. See our response RC1-5 below for more on this point. If the editor allows a revision, we would be happy to provide further discussion of these premises.

Does this response occur in other ecosystems or are the only observations from the Costa Rica site? The introduction gave several examples of how different ecosystems and different individuals within a given ecosystem respond differently to N and P fertilization. Are there any patterns that emerge across ecosystems? If this response is specific to a single or a small number of sites, why should it be represented in TBMs?

(Response RC1-3): Our observations are only from the Costa Rica site. Cunha et al. (2022) carried out a fertilization experiment on old-growth Amazonian sites and also found that relative and absolute allocation to fine roots increased under P fertilization. In another Amazonian fertilization experiment, Lugli et al. (2021) reported that absolute fine root production increased, but did not comment on relative allocation to fine roots. Given these studies, we think that this response is relevant to TBMs because it has been seen in a large proportion of tropical studies, even if the absolute number of studies with leaf/wood/root productivity measured separately is relatively small. In addition, the model simulations that we carried out can be useful for inspiring future experimental design and field work.

Previous global meta-analysis (Yuan and Chen 2012) showed that fine root production increased with P fertilization, but relative allocation to fine roots decreased. The (global) decrease in relative fine root production contrasts with the tropical forest studies of Waring et al. (2019) and Cunha et al. (2022). However, Yuan and Chen (2012) did note substantial variation across ecosystems. One potential confounding problem in making comparisons is that different experiments used different fertilizer amounts, affecting the fertilization response (Hou et al. 2020). Thus, we are reluctant to speculate too much on global patterns.

Have empirical studies indicated that this is important for larger C fluxes? This is somewhat touched on in the Discussion but its prevalence was not clear.

(Response RC1-4): We are not aware of its relevance for larger C fluxes on the basis of previous empirical studies. However, our expectation is that its relevance should be of interest (Line 69-73). Related to this point, we argue that some of our figures should stimulate hypotheses and prompt more field work in the future.

2. How do other factors interact to determine relative allocation to roots? Water and nitrogen should play important roles as well. Is it valid to only focus on P (especially given that the results suggest that it is increased water uptake that seemed to drive AGB)?

(Response RC1-5): We focused the relationship between soil P and relative allocation to fine roots on the basis of a previous fertilization experiment (Waring et al. 2019). That experiment showed that relative allocation to fine roots was impacted by P fertilization but not N fertilization. Furthermore, the experiment spanned years of relatively high and low rainfall, and thus allowed us to see how the actual fine root-to-leaf productivity ratio varied with rainfall. Specifically, rainfall in 2015 was 628 mm, rainfall in 2016 was 1754 mm, and rainfall in 2017 was 2050 mm. The observed leaf:fine root productivity ratio did not exhibit a clear relationship with rainfall. Averaged across treatments, this productivity ratio was 0.37 in 2015, 0.44 in 2016, and 0.24 in 2017. We also looked at the leaf:fine root productivity ratio by treatment. In all cases, this productivity ratio was intermediate in 2015, largest in 2016, and smallest in 2017.

Because the observations lacked a clear correlation with rainfall, we decided not to focus on the impacts of water on the leaf:fine root productivity ratio.

We acknowledge that our original manuscript did draw a correlation between simulated transpiration and simulated aboveground biomass (AGB) (Figs. 7a, 8). However, prompted by this comment as well as a comment from a community reviewer (response CC1-11), we carried out more detailed analysis of this point. Our current thinking is that the correlation between transpiration and AGB is causal. Instead, the higher transpiration in the "pos" schemes occurred because the "pos" schemes had larger leaf area. Transpiration per unit leaf area was similar in all schemes.

We therefore decided to look more deeply into why the "pos" schemes had relatively large AGB in Fig. 7a. Following a community comment (response CC1-9), we carried out additional simulations spanning a wider range of values for our allocation parameters $a$ and $b$ (these parameters are defined in Eq. 1 of the original manuscript). For each choice of $a$ and $b$, AGB at the end of 30 years is shown here (we only show the average of the plots without P fertilization, thus corresponding with the manuscript's Fig. 7a):

[Figure]

*Simulated AGB in different parameterizations with wider range of a and b.*

This fuller analysis shows that AGB does not simply increase with $b$. To explain this pattern, we hypothesized that the highest AGB would occur when the relative acquisition of C:P was optimal. We therefore computed the ratio of nonstructural C to nonstructural P and plotted it against AGB:

[Figure]

*The relationship between simulated AGB and C:P in nonstructural pool for different parameterizations.*

The highest nonstructural C:P ratios corresponded most closely to leaf C:P (which was fixed at 600 g C (g P)$^{-1}$) and to the highest simulated AGB values. This result confirms the hypothesis.

Based on this result, it could be argued that our allocation rule for relative allocation to fine roots should be targeted to the nonstructural C:P ratio rather than soil P. We indeed think that such a scheme would work well in the unfertilized plots; however, it would not capture the P fertilization effect (increased relative fine root production with increased soil P).

Additionally, I would assume that the role of other plant mechanisms to increase P uptake would be important as well, such as phosphatase synthesis and arbuscular mycorrhizae. These are likely intricately linked to fine root biomass in real ecosystems. While these do not necessarily need to be examined or modelled, they should be at least recognized in the experimental setup and discussion.

(Response RC1-6) We agree that other plant mechanisms to increase P uptake could be important as well. In a revision, we would be happy to discuss additional plant mechanisms to increase P uptake, including phosphatase synthesis (Liu et al. 2015; Kong et al. 2016; Lugli et al. 2020) and arbuscular mycorrhizae (Hodge 2004; Comas et al. 2014; Eissenstat et al. 2015; Liu et al. 2015; Kong et al. 2016; Ma et al. 2018). For example, root phosphatases hydrolyze organic P-containing compounds and releasing inorganic P that is absorbable by roots; mycorrhizal associations are even more effective by enlarging the root absorbing surface per unit cost; both mechanisms provide additional P sources. Plants adaptively adjust their traits or metabolic processes in terms of effective P acquisition (Raven et al. 2018; Han et al. 2021; Aoyagi et al. 2022), and diverse P acquisition strategies are being evaluated from observations (Reichert et al. 2022).

Additionally, flexible stoichiometry could be important. How have other models approached these phenomena?

(Response RC1-7): Current models approach stoichiometry differently. Some have fixed stoichiometries (e.g. JSBACH, Goll et al., 2012; CLM-CNP, Yang et al., 2014; JULES-CNP, Nakhavali et al., 2022). Some account for stoichiometric flexibility by prescribing ranges for each pool based on empirical studies (e.g. CASACNP, Wang et al., 2010; ORCHIDEE-CNP, Goll et al., 2017; QUINCY, Thum et al., 2019). ED2 has fixed stoichiometries in structural pools but non-fixed stoichiometries in non-structural pools (Line 131 in the original manuscript). Models adjusting relative allocation of new growth to fine roots mostly apply an idea that new growth is scaled by minimum of N and P stress scaling factor, rendering increased fine root production when P demand exceeds supply (for example, when the ratio of nonstructural C:P greatly exceeds the ratio present in plant tissues). By contrast, we parameterized the model only from the perspective of supply (soil P). We would be happy to provide more discussion on this point in a revision.

3. It was not made clear which PFT was being studied in these experiments. Were there multiple PFTs? Given that this is a dry tropical forest, do deciduousness and phenology play a role here? How could these results differ between tropical dry forests and tropical moist forests? Have similar experiments been conducted in tropical moist forests?

(Response RC1-8): The model included eight PFTs. Species are assigned to a PFT on the basis of three traits: wood density, specific leaf area, and legume/non-legume status. The binning of species into PFTs is discussed in Xu et al. (2016) and Medvigy et al. (2019). Deciduousness is not pre-assigned, but is rather an emergent model outcome (Xu et al. 2016).

In this dry tropical forest, intra-annual variability of precipitation does influence plant phenology: new leaves are produced in April and May and shed between January and March; stems did not grow during the dry season (Waring et al., 2019). Therefore, the fertilization was conducted only during wet season; measurements of leaf/wood/fine root production were conducted differently to accommodate seasonality (Waring et al., 2019). Our model simulations and the calculation of productivity were in accordance with this experiment.

Analogous experiments have been conducted in tropical moist forests (please see response RC1-3).

4. Using different statistical analyses for leaf, wood, and root due to patterns that emerged from the observation-based data may not be the best approach. It would be a more direct comparison to use the same statistical analyses for each tissue because the biases in the empirical observations may not be present in the model outputs. Figure 4 is a central figure but it is unclear whether it shows only the control treatment or an average across treatments. Regardless, this analysis should be conducted for each fertilization treatment independently given that the premise of the study is that fertilization treatment influences relative allocation. Furthermore, are the temporal trends important here given that the same amount of fertilizer was applied each year and the experiment was only 3 years long? Given that the primary focus is the difference between tissues rather than the difference between years, it may make more sense to aggregate across years for each tissue / treatment.

(Response RC1-9): We thank the reviewer for these suggestions. We re-analyzed these results largely following this advice. First, we averaged over years. The main drawback to this averaging is that we lose some statistical power because all t-tests would need to be done on only four replicates. However, when we carried out this analysis, we found that the impact on our results was small (see below). We also performed statistical tests by treatment. An advantage of following this approach is that we no longer had concerns about mixing samples that were drawn from different distributions. We therefore performed t-tests for leaf, wood, and fine root productivity. We no longer found it necessary to do the regression on fine root productivity. As a result, productivity of leaves, wood and fine roots were analyzed in exactly the same way.

First we looked at leaf productivity. We carried out t-tests to determine whether there were significant differences between simulations and observations. In response to a community comment (please see response CC1-9), we increased the number of parameterizations that we tested. If, for a particular treatment, the p value was less than 0.05 (the threshold which we took to indicate a significant different between the simulations and the observations), we indicated that in the table below. "C" means the control plots were significantly different, "N" means the +N plots were significantly different, "P" means the +P plots were significantly different, and "B" indicates that the +NP plots were significantly different. If an entry is empty, it means that no significant differences were found for any treatment. Here are our results for leaf productivity:

| LEAF PRODUCTIVITY | a=0 | a=0.1 | a=0.2 | a=0.3 | a=0.4 | a=0.5 | a=0.6 | a=0.7 | a=0.8 |
|---|---|---|---|---|---|---|---|---|---|
| b = -60 | | | | | | | | | |
| b = -40 | | | | | | | | | |
| b = -20 | | | | | | | | | |
| b = 0 | | | | | | | | | |
| b = 20 | | | | | | | | | |
| b = 40 | | | | | | | | | |
| b = 60 | P | P | | | | | | | |

As in our original manuscript, almost all model parameterizations successfully predicted leaf productivity.

We then looked at wood productivity:

| WOOD PRODUCTIVITY | a=0 | a=0.1 | a=0.2 | a=0.3 | a=0.4 | a=0.5 | a=0.6 | a=0.7 | a=0.8 |
|---|---|---|---|---|---|---|---|---|---|
| b = -60 | P | P | P | P | P | | | | |
| b = -40 | P | P | P | P | P | | | | |
| b = -20 | P | P | P | P | P | P | | | |
| b = 0 | P | P | P | P | P | P | P | | |
| b = 20 | P | P | P | | | | | | |
| b = 40 | | | | | | | | | |
| b = 60 | | | | | | | | | |

Many parameterizations, especially with $b < 0$, $b = 0$, or relatively small $a$, did not predict wood productivity in the +P treatment. Parameterizations with $b > 0$ were mostly successful in all treatments. Again, this result is consistent with our original manuscript. However, here we have more information as compared to our original manuscript: we see that the model only ever had trouble with the +P treatment, and not the +NP treatment.

Finally, we looked at fine root productivity:

| FINE ROOT PRODUCTIVITY | a=0 | a=0.1 | a=0.2 | a=0.3 | a=0.4 | a=0.5 | a=0.6 | a=0.7 | a=0.8 |
|---|---|---|---|---|---|---|---|---|---|
| b = -60 | C,N,P,B | C,N,P,B | C,N,P,B | C,N,P,B | C,N,P,B | P,B | P,B | C,P | C,P |
| b = -40 | C,N,P,B | C,N,P,B | C,N,P,B | C,N,P,B | P,B | P,B | P,B | C,P | C |
| b = -20 | C,N,P,B | C,N,P,B | C,N,P,B | C,N,P,B | P,B | P,B | P | C | C |
| b = 0 | C,N,P,B | C,N,P,B | C,N,P,B | C,N,P,B | P,B | P | P | C | C |
| b = 20 | C,N,P,B | C,N,P,B | C,N,P | N,P | | | C | C | C,B |
| b = 40 | C,N,P | C,N | N | | | C,B | C,P,B | C,P,B | C,N,P,B |
| b = 60 | C,N | N | B | P,B | C,P,B | C,P,B | C,P,B | C,P,B | C,N,P,B |

Only four parameterizations, all with $b = 20$ or $b = 40$, successfully predicted fine root productivity in all treatments. These four parameterizations also successfully predicted leaf and wood productivity in all treatments.

Again, the drawback of this approach is that is reduced the number of samples: each t-test was done with only four replicates. However, our new results are similar to what we reported in the original manuscript. As an additional test, we tried aggregating control and +N, and comparing that to the aggregation of +P and +NP. This procedure doubles the number of replicates. It gave very similar results (not reported here).

5. Is a 2 year spinup sufficient? Shouldn't the spinup be run until an equilibrium is established?

(Response RC1-10): We did not want to spin up the model to equilibrium because the actual forest is only about 30 years old and is not in equilibrium. Rather, we used the observed stand structure and composition and observed soil nutrient status to initialize the model. Our two-year spin-up was done to initialize soil water. For further rationale, see Xu et al. (2016). We would be able to provide more justification of the spinup in a revision.

Technical correction

Line 81 "While models have rarely be validated on these time scales" I would argue that models are often evaluated over the past several decades (1960s to present).

Here we wanted to emphasize that nutrient fertilization experiments have usually lasted for a few years and hence the lack of long-term benchmarks for model validation. We can rephrase it to make it clearer.

Production units should be kg m-2 yr-1.

Thanks for pointing this out. We can correct the units in a revision.

Table 3 is challenging to interpret. Could this be transformed into a figure?

Yes, we can make such a figure. See the figures given in response RC1-5 for examples of the potential format.

Include other parameterizations in Figure 5 (additional panels).

As discussed above, this figure will be eliminated in the revision.

Include other treatments in Figure 8 (additional panels).

As discussed above, we plan on eliminating this figure.

Figure 6: Clarify if this is averaged across treatments or if this is the control treatment only.

This is the average across treatments. We can clarify it in the caption.

**References**

[revised manuscript text omitted]

---

## Author Comment (AC2)

Dear Dr. Katrin Fleischer,

Thank you for the constructive comments. We copied your comments below in blue font; our labeled responses are in ordinary black font. We have labeled our responses for ease in cross-referencing. Responses to your comments are labelled "CC1" and responses to Reviewer 1 are labeled "RC1". We have not yet received responses from a second reviewer. We welcome further discussion during the discussion period.

The premise of the study:

The authors confront a dynamic vegetation model with experimental data from a nutrient fertilization experiment in Costa Rica and assess the validity and consequences of alternative root allocation parameterization. The overarching question the authors address is how root allocation responds to changes in nutrient availability, which is a very valid question to pursue. Evaluating models and their underlying assumptions with direct field observations and experiments is necessary to advance our understanding and gain confidence in model predictions. I appreciate the effort the authors have undertaken with the demographic model ED2, and their joint effort with experimental findings and expertise to advance root dynamics and their interactions with nutrient availability. However, I currently cannot recommend publishing the manuscript due to some concerns.

(Response CC1-1) Thank you for these positive comments, and for the many great suggestions that follow. In a revision, we think that we would revise how we state our overarching questions in order to better align them with our results and discussion. Specifically, we are trying to answer: (a) What model assumptions enable or prevent the model from being consistent with the observations? (b) How can we use the model to generate hypotheses on the longer-term consequences of allocation assumptions?

Main concern:

1. The ecological underpinnings of the chosen parameter sets in regard to fine root allocation are not sufficiently clear. This includes the ecological theory to justify choosing the parameter sets, the discussion of the consequences of these parameter sets, and placing the chosen parameter set and findings within the literature and previous modeling efforts.

(Response CC1-2) Thank you for this comment. Reviewer 1 brought up similar concerns. Please see our responses RC1-2, RC1-3, and RC1-7.

2. The study covers short-term and long-term process effects of the different parameterizations, whereby we can expect different outcomes. The experiment finds that fine root production increases with fertilization.

(Response CC1-3) We agree with this comment.

That does not necessarily mean that higher fertile sites will be characterized by higher fine root production.

(Response CC1-4) For clarification, the 3-year experiment showed that P fertilization did increase fine root production (Waring et al. 2019). Whether this response will persist in the longer term is currently unknown, but we hope to eventually have the experimental data to test this point (lines 365-366 in the original manuscript).

In addition, we are open to the idea that allocation responses to P fertilization on a three-year time scale might differ from correlation analysis (e.g., correlation of soil P with fine root productivity across a strong fertility gradient). Individuals growing on naturally high-P sites, for example, may have had sufficient time to acclimate to their local environment. Also, on decadal to multidecadal time scales, forest demand for P may change as a function of forest age or as species turn over, and such changes can affect relative allocation. We can discuss the distinction between fertilization experiments and correlation analysis in a revision.

> The others touch on these aspects of timing in the discussion, and the significance of this difference is crucial for the study, however insufficient emphasis is placed on this in working out the premise of the study and discussing the findings.

(Response CC1-5) We are not sure that we fully understood this comment. Which "others" are being referred to here? We are very open to the idea that short term (3-year) and long term (30-year) responses might be different (lines 77-83), and that is indeed part of the reason why we carried out 30-year simulations.

We note that in our 30-year model simulations, successional changes in forest structure, composition, and function are allowed to occur. But the allocation rule remains fixed (no acclimation). We think it is interesting that the parameterizations that matched the 3-year observations most closely simulated the least AGB accumulation on 30-year time scales with P fertilization. This model result does beg the hypothesis that some acclimation would occur on 30 year time scales. This idea can mathematically be incorporated into the model by having relative fine root allocation be a saturating function of soil P rather than a linear function. Such a rule would help to prevent over-allocation to fine roots under the very high soil P concentrations associated with P fertilization. We hope that these remarks speak to your concern and we would be happy to summarize them in our revision.

> Associated with this, the experiment takes place in a young forest stand, the implications of this deserve more discussion.

(Response CC1-6) We agree that forest age could be an important variable related to relative allocation to fine roots. Forests of different ages would be expected to have different composition, structure, and demographic rates. It is interesting that the pattern seen in the relatively young forest of our analysis was similar to the pattern seen in the old growth forests studied by Cunha et al. (2022). In a revision, we are happy to comment on this point.

> 3. Direct nutrient acquisition via fine roots is only one of several possible mechanisms of how plants can acquire nutrients. The authors mention that the trees are associated with arbuscular mycorrhizae but the implications of this in regard to the outcome of the experiment, and how this might have affected the model performance are not addressed.

(Response CC1-7) Please see our response RC1-7. Waring et al. (2019) found root colonization by mycorrhizal fungi did not vary among nutrient treatments. In the manuscript (Line 386-389)

we discussed the possibility of increased root production but reduced allocation to mycorrhizae, which together might show overall decreased belowground allocation.

Additional comments methodology:

The chosen parameter sets reflect negative, constant, and positive relationships between fine root production and soil P. The negative one would reflect the resource-dependent parameterization, however not via internal plant demand and supply (as commonly done) but depending on external P supply, in a linear fashion. This is quite different from any of the previous model approaches, and the parameterizations are all based on a linear relationship between soluble soil P and root allocation. A discussion of the ecological underpinning of this model and the parameterization approach would be helpful. The authors touch upon the ecological theory they are addressing only in the discussion part.

(Response CC1-8) Please see our response CC1-2.

Parameters a and b are coordinated to yield a similar 0.3 root-to-shoot ratio in control plots, but to different settings in fertilized plots. The reason for not testing different parameterization settings in the control plot was not clear to me, it would be helpful if the authors could elaborate on this.

(Response CC1-9) Thank you for this particularly helpful comment. We carried out more simulations so that we more densely sampled parameter space. Some of the main results are presented in our responses RC1-5 (30-year time scale) and RC1-9 (3-year validation).

The allocation process as a whole in ED2 is not sufficiently explained. The calculation of the daily leaf and root allocation, and the allometric equations to determine maximum leaf and root allocation, need to be described and included in the discussion. Similarly, the PFTs that are modeled and their parameterizations are not described.

(Response CC1-10) We originally did not present this information because it can be found in other publications (Longo et al. 2019a, Medvigy et al. 2019). However, we would be happy to also provide this information in a revised version of this manuscript. See also response RC1-8.

Additional comments results:

One positive parameterization was the only one that agreed well with experimental observations over the 3-year period. Positive parameterizations led to the highest AGB over 30 years, however, not due to alleviation of nutrient limitation but water stress. It would be helpful if the reader would understand how that occurred process-wise. What was the actual allocation to fine roots? And why did that not alleviate nutrient stress?

(Response CC1-11) For a process-level explanation, see response RC1-5. The actual allocation to fine roots is shown on the next page, together with other biomass pools.

[Figure]

Similarly, with P fertilization, positive parameterizations led to too many fine roots over the 30-year time scale. As is described in the abstract as the main finding. Does that not indicate that the chosen parameterization works well for the short term but, importantly, not well in the long term? Is such an assumption about root allocation advisable then? It would be helpful if the authors go beyond "experiments need to measure leaf, root and wood production" in their main take-away for the paper, and rather elaborate on recommendations for model development.

(Response CC1-12) We think that the parameterization works well in the short term in the sense that it yields simulation results that are consistent with observations. We do not know how well the parameterization would work in the long term because we do not have long-term observations to compare the simulations to. Thus, we see a strong need for longer-term experiments exactly in order to make such assessments. We did not mean to imply that the positive parameterization is a "chosen" parameterization, and all of our results displayed multiple parameterizations. We can try to be more clear about this point in a revision.

We offer several additional thoughts spurred by this comment, which we can work into our Discussion: (1) It could be that a saturating response of relative fine root allocation to soil P (rather than a linear response) would maintain the model's capability of simulating the 3-year observations and also maximize AGB under fertilized and unfertilized conditions. But this would be a more complicated parameterization than the one we employed, and so we did not favor it. (2) It could be that P fertilization over 30 years would expose trees to soil P concentrations that are well outside the natural range. In such a novel environment, the response may well be maladaptive. (3) Under "normal" unfertilized conditions, there does not appear to be any problem with positive parameterizations (Fig. 7a in the original manuscript). As noted in response RC1-5, a scheme that employed a target nonstructural C:P might yield a similar result (but again, such a scheme is unlikely able to simulate the increase in fine root production observed to occur with P fertilization).

We think that model development and experimental work should be coupled. Given our modeling results, we think that the natural next step should be more experiments. Additionally, testing of different parameterizations should be done across a gradient as in the Amazon where data are available.

Given that the experiment took place in a tropical dry forest, the anticipated interactions between soil fertility and water stress are intriguing and it would be helpful if the authors elaborate on the alleviation of water stress in the 30-year simulation that has been touched upon.

(Response CC1-13) Please see our response RC1-5.

Maybe the agreement at the 3-year time scale is not sufficient to evaluate.

(Response CC1-14) We are not exactly sure what is meant here. A three-year time scale is not uncommon in ecosystem experiments. Further, the model represents processes on time scales ranging from seconds to centuries. Ideally, we would constrain the model over a range of time scales (for example, see Trugman et al. 2016; Longo et al. 2019b). Because the Waring et al. (2019)'s fertilization occurs within the range of time scales that the model simulates, we do think

it is appropriate to validate the model on a three year time scale. Note that the model has been validated on somewhat longer time scales at nearby unfertilized plots (Medvigy et al. 2019).

Can the control plots not also be used to evaluate the parameterization, representing rather long-term dynamics? Should the parametrizations not be tested on the control and experimental plots, equally?

(Response CC1-15) We agree that this would be helpful. Please see our response RC1-9. These new results can be included in the revision.

The authors state that ED2 has been validated before. Did the constant parameterization work well before? Did ED2 employ a resource-dependent allocation scheme before? It would be helpful to outline which allocation approaches agreed well with observations and which did not in previous studies with ED2.

(Response CC1-16) ED2 has been validated at a range of ecosystems and for a range of quantities (Medvigy et al. 2009; Medvigy and Moorcroft 2012; Trugman et al. 2016; Xu et al. 2016; Levy-Varon et al. 2019; Longo et al. 2019b; Medvigy et al. 2019). The "constant parameterization" is the one built in to the basic model code, and has never been contrasted with a resource-dependent allocation scheme before. Generally speaking, the model has only been validated for aboveground quantities like AGB production, leaf production, and eddy fluxes. We show here that many parameterizations lead to reasonable wood and leaf production (as the model has been previously validated for), but few lead to reasonable fine root production (as the model has rarely or never been validated before).

The resource limitation theory postulates plant allocation is adjusted to acquire the most limiting resource. Many ecosystem models to date, adopt a resource-dependent allocation scheme, so they would predict that more roots would be produced if soil nutrients were limited, irrespective of the return. The authors find that the experimental results contradict this hypothesis. I believe this apparent contradiction might be a time issue, so that roots grow to acquire the soluble P, while root allocation would decrease once P limitation is alleviated. It might also have to do with mycorrhizal interactions, previous outsourcing of phosphorus acquisition to mycorrhizae now becomes less beneficial with increased nutrient supply and plants switch to "do-it-yourself". Even if we cannot be certain about any of these hypotheses, a discussion thereof would be helpful.

(Response CC1-17) Thank you for these points. We think that they have largely been addressed in our author comments (different ecosystem models were mentioned in RC1-7, timing issues in response CC1-5 and CC1-12, and mycorrhizal colonizations in CC1-7). We would be happy to revise our manuscript to include discussion of these points.

Line comments:

1. 336 please specify: allocation parameterization sensitive to external nutrient availability.

(Response CC1-18) Thanks for pointing this out. We'll make it clear in revision.

2. 383- 341 what is the implication of that finding?

(Response CC1-19) Is there a typo in the line numbers specified here? We were not sure which lines the reviewer referred to.

   3. 344 the model is microbial-explicit. That is an important aspect of the model. It would be helpful if this is included in the manuscript and discussed.

(Response CC1-20) As suggested by the last comment on methodology, we can include model details about microbial mechanisms of soil organic matter decomposition in a revision.

   4. 365 For future efforts, the authors can consider evaluating the model and different parameter sets at different locations in the tropical biome, and including an evaluation along a soil fertility gradient, as well as experimental changes. The combination of both would be helpful to discern short-term and long-term effects.

(Response CC1-21) Thanks for the suggestion and please see also our response CC1-4. We can add this to the discussion.

   5. 369 The experiment near Manaus found increased primary productivity in response to fertilization, indicating that production was limited by phosphorus. To my understanding, this finding is still consistent with the resource limitation hypothesis, since plants allocated carbon to roots to acquire the nutrients that they were in demand of.

(Response CC1-22) Of concern to us is that relative – and not just absolute – allocation to fine root increased in Cunha et al. (2022) in response to P fertilization.

   6. 375 As the authors note, roots are there for acquiring multiple resources at once. A discussion of these interactions would be helpful. The analogy to rain roots is interesting, the authors could elaborate here. The root production after fertilization could potentially be a similar short-term effect to acquire the limiting resource.

(Response CC1-23) We could briefly elaborate on these points in a revision. Regarding time scales, please see our responses CC1-4, CC1-5, and CC1-12.

   7. 379 The paragraph on field observation is helpful, however, root stock and root production seem to be mixed up. Since the model evaluation deals with root production, the authors could elaborate on those aspects of the measurements.

(Response CC1-24) We could not find the text where we mixed up root stock and root production.

   8. 387 It would be helpful if the authors elaborate on the interactions with mycorrhizae, here, and the implications for this study.

(Response CC1-25) We can comment on this in a revision. Please see response CC1-7.

   9. 389 The "supply-limited" hypothesis, which is the basis of this study, should be introduced earlier and placed in context with the alternative hypotheses.

(Response CC1-26) Thanks for the suggestion. We can introduce this concept earlier when setting the premises of this study in a revision.

10. 390 This section could benefit if it discussed what has been learned from this and the previous approaches.

(Response CC1-27) We agree that more can be said here. Some important points include: (1) the sensitivity on one time scale (3 years) can be different from the sensitivity on another time scale (30 years). (2) Different measures of productivity differ in their sensitivity. Just because one measure of productivity is sensitive to a particular parameter does not mean that the model is generally sensitive to that parameter. (3) For some measures of productivity, model sensitivity strongly depends on the nutrient context (that is, whether or not P fertilization is applied).

11. 398 Similarly, in this section, it would be helpful if the authors go beyond the summary of the results here and discuss the implications of these findings. See the main comments above.

(Response CC1-28) Thanks for the suggestion. We disagree with the characterization of this text as a "summary of results"; rather, the text is more of an "interpretation" of results, and so it should be appropriate for a discussion section. The text discusses implications in other places, especially in Section 5, but also to a certain extent in Section 4.4 and other locations.

12. 424 Here it is crucial to discuss that resource-dependent model approaches are not exactly comparable to the negative parameterization applied in this study. See the main comments above.

(Response CC1-29) We appreciate this suggestion and can follow through in a revision.

**References**

Cunha, H.F.V., Andersen, K.M., Lugli, L.F., Santana, F.D., Aleixo, I.F., Moraes, A.M., Garcia, S., Di Ponzio, R., Mendoza, E.O., Brum, B. and Rosa, J.S., 2022. Direct evidence for phosphorus limitation on Amazon forest productivity. *Nature*, *608*(7923), pp.558-562.

Levy-Varon, J.H., Batterman, S.A., Medvigy, D., Xu, X., Hall, J.S., van Breugel, M. and Hedin, L.O., 2019. Tropical carbon sink accelerated by symbiotic dinitrogen fixation. *Nature communications*, *10*(1), p.5637.

Longo, M., Knox, R.G., Medvigy, D.M., Levine, N.M., Dietze, M.C., Kim, Y., Swann, A.L., Zhang, K., Rollinson, C.R., Bras, R.L., Wofsy, S.C. and P.R. Moorcroft 2019a. The biophysics, ecology, and biogeochemistry of functionally diverse, vertically and horizontally heterogeneous ecosystems: The Ecosystem Demography model, version 2.2–Part 1: Model description. *Geoscientific Model Development*, *12*(10), pp.4309-4346.

Longo, M., Knox, R.G., Levine, N.M., Swann, A.L., Medvigy, D.M., Dietze, M.C., Kim, Y., Zhang, K., Bonal, D., Burban, B., Camargo, P.B., Hayek, M.N., Saleska, S.R., da Silva, R., Rollinson, Bras, R.L., Wofsy, S.C. and P.R. Moorcroft, 2019b. The biophysics, ecology, and biogeochemistry of functionally diverse, vertically and horizontally heterogeneous ecosystems: The Ecosystem Demography model, version 2.2–Part 2: Model evaluation for tropical South America. *Geoscientific Model Development*, *12*(10), pp.4347-4374.

Medvigy, D. and Moorcroft, P.R., 2012. Predicting ecosystem dynamics at regional scales: an evaluation of a terrestrial biosphere model for the forests of northeastern North America. *Philosophical Transactions of the Royal Society B: Biological Sciences*, *367*(1586), pp.222-235.

Medvigy, D., Wofsy, S.C., Munger, J.W., Hollinger, D.Y. and Moorcroft, P.R., 2009. Mechanistic scaling of ecosystem function and dynamics in space and time: Ecosystem Demography model version 2. *Journal of Geophysical Research: Biogeosciences*, *114*, G01002.

Medvigy, D., Wang, G., Zhu, Q., Riley, W.J., Trierweiler, A.M., Waring, B.G., Xu, X. and Powers, J.S., 2019. Observed variation in soil properties can drive large variation in modelled forest functioning and composition during tropical forest secondary succession. *New Phytologist*, *223*(4), pp.1820-1833.

Trugman, A.T., Fenton, N.J., Bergeron, Y., Xu, X., Welp, L.R. and Medvigy, D., 2016. Climate, soil organic layer, and nitrogen jointly drive forest development after fire in the North American boreal zone. *Journal of Advances in Modeling Earth Systems*, *8*(3), pp.1180-1209.

Waring, B.G., Pérez-Aviles, D., Murray, J.G. and Powers, J.S., 2019. Plant community responses to stand-level nutrient fertilization in a secondary tropical dry forest. *Ecology*, *100*(6), p.e02691.

Xu, X., Medvigy, D., Powers, J.S., Becknell, J.M. and Guan, K., 2016. Diversity in plant hydraulic traits explains seasonal and inter-annual variations of vegetation dynamics in seasonally dry tropical forests. *New Phytologist*, *212*(1), pp.80-95.

---

## Author Comment (AC3)

Dear Reviewer,

Thank you for the constructive comments. We copied your comments below in blue font; our responses are in ordinary black font. We have labeled our responses for ease in cross-referencing. Responses to your comments are labeled "RC2". Cross-references to our responses to Reviewer 1 are labeled with "RC1" and cross-references to our responses to a community reviewer are labeled with "CC1".

Shuyue Li et al., conducted a modeling study to investigate how plant allocation in response to nutrient fertilization. They used nutrient enabled ED2 model with various different parameterizations on biomass allocation under control and fertilized conditions over tropical dry forest. Data from an fertilization experiment at Costa Rica forest was used for model comparison and validation. The paper is well organized and presentation is smooth. Below I have a few suggestions and comments.

(Response RC2-1): Thank you for your generally positive review and many great suggestions and comments. Please see our responses to each of them below.

1. introduction, first paragraph needs to be improved, Nutrient availability could affect plant activity in many different ways. The most relevant (to this paper) way is through mediation C/N/P allocation and biomass construction. However, the first paragraph try to explain how nutrient availability could affect plant response to CO2 enrichment, which is not much relevant here.

(Response RC2-2): In the first paragraph of the manuscript, we wanted to provide the overall context for this study: more process-level understanding of nutrient limitation is essential for reliable prediction on the primary production of terrestrial ecosystem under future environmental change. Yet we think that this is a fair critique. We will plan to revise this first paragraph to be briefer on the general context and to better introduce the idea that C/N/P stoichiometry can be critical for biomass construction and allocation.

2. introduction, paragraph 2 and 3 provide a nice summary of many fertilization experiments for tropical trees. However, each fertilization experiment was discussed individually. I would suggest adding some discussion about why and how experiment results differ from one another to improve the coherence of the summary.

(Response RC2-3): In Paragraph 2 and 3, we mentioned existing nutrient fertilization experiments and discussed them individually mainly because of little effect was found on stand level but large variability appeared across studies. But we also agree that adding a few more sentences of synthesis would improve the coherence of the discussion. We would be happy to follow the reviewer's suggestion in a revision.

3. introduction, paragraph 5 and 6 highlight the need to investigate and improve the allocation scheme under long-term fertilization for current generation CNP models. In this case, a survey of allocation schemes used by current generation models are necessary, for example some models assume constant allocation, some assume multiple

resource coordination, some are based on carbon cost …Besides the seven CNP models mentioned in this section, two more recent global CNP models are:FUN-CNP: Braghiere, R.K., Fisher, J.B., Allen, K., Brzostek, E., Shi, M., Yang, X., Ricciuto, D.M., Fisher, R.A., Zhu, Q. and Phillips, R.P., 2022. Modeling global carbon costs of plant nitrogen and phosphorus acquisition. Journal of Advances in modeling earth systems, 14(8), 2022MS003204. ELM-CNP: Zhu, Q., Riley, W.J., Tang, J., Collier, N., Hoffman, F.M., Yang, X. and Bisht, G., 2019. Representing nitrogen, phosphorus, and carbon interactions in the E3SM land model: Development and global benchmarking. Journal of Advances in Modeling Earth Systems, 11(7), 2238-2258.

(Response RC2-4): We agree with this suggestion. We can provide further discussion on the allocation schemes of current generation CNP models in Section 4.4. We can also add an overview of various allocation schemes in Introduction as suggested. We appreciate the reviewer for bringing up these other CNP models. ELM-CNP (Zhu et al. 2019) applies a resources-dependent allocation scheme developed by Friedlingstein et al. (1999), analogous to our "neg" parameterizations; Braghiere et al. (2022) integrated the most recent version FUN3.0 with ELM, modulating plant nutrient uptake from multiple pathways by optimizing carbon cost, but did not illustrate how this strategy might affect new-growth allocation; moreover, neither of them discussed the effects of nutrient availability on relative allocation to leaves, wood and fine roots. We are happy to include the discussion on additional CNP models in a revision.

4. section 2.3, r2l is a function of soil P concentration (psol), I wonder mathematically will this equation lead to huge variability of r2l parameters especially at the time when fertilizers were applied. Maybe showing a figure of r2l during the 3 years of fertilization experiment will help to clarify this.

(Response RC2-5): The variability of the r2l parameter depends on treatment and parameterization. Without P fertilization, there is some seasonal variability in r2l, but it is relatively small. The variability is largest under P fertilization with the "pos" parameterizations, where r2l ranges mostly from 0.4 to 1.0. Despite this variability, the "pos" parameterization consistently leads to larger r2l under P fertilization than the "const" or "neg" parameterizations. We can include such a figure (see below) and some discussion in a revision.

[Figure]

5. section 2.4.1. Vegetation and soil are both initialized with in situ observations, rather than being determined by long-term spinup. Such approach often time will result in an dis-equilibrate vegetation and soil processes. Therefore, after initialization the vegetation and soil states will quickly changes towards quasi-equilibrate conditions, which could be largely different from the initialized conditions. I wonder if the re-equilibration also occur in ED2, how long does it re-equilibrate, and how that affect fertilization results?

(Response RC2-6): This question is difficult to answer because our study system is a 30-year secondary forest and the actual forest is not equilibrated yet. Because the actual forest is still unequilibrated, we did not intend to spin up the model to equilibrium. The two year spinup was sufficient to initialize soil water (see also in Response RC1-10). Despite the short spinup, the model has been shown to simulate aboveground biomass reasonably well (Medvigy et al. 2019) and no obvious transients in leaf area index have been observed (Xu et al. 2016). The plant-available soil nutrient pools (Figure 2 in the original manuscript) also do not exhibit much of a

transient. The slowly varying soil pools were initialized from observations and were not expected to change much over the course of this study because they are slowly varying.

6. section 2.4.1. It was mentioned that fine root production was evaluated with linear regression, however, it also mentioned linear regression was not appropriate because there existed only three years of data. Here needs more clarification.

(Response RC2-7): Thank you for pointing this out. Reviewer 1 came up with a similar concern. Please see our detailed responses in Response RC1-9. In short, we reanalyzed our simulation results by first averaging over years, and then performing t-test for leaf, wood, and fine root productivity by treatment, to determine whether there were significant differences between simulations and observations. By doing so we no longer need linear regression to evaluate model performance on fine root production. We will use this fuller version of analysis in revision.

7. Figure 2, most of the simulated variability of NO3, NH4 already exist in control run (solid blue lines), it doesn't look like there were sudden increase of NH4 or NO3 right after the N fertilization. Also, it will be helpful, if the fertilization date could be marked on the x-axis.

(Response RC2-8): We agree with this comment, as mentioned in original manuscript (Line 235). This is likely because leaching is an important pathway for N loss, especially during wet seasons when fertilizers were applied. We are happy to make modifications on Figure 2 to display approximate fertilization time.

8. Section 4.1. It's still not clear to me which parametrization is the best. It was stated that "only one of the 13 parameterizations that we tested was able to simultaneously simulate leaf, wood and fine root (missing word) production consistent with the observations". Here, the screensful parameterization needs to be highlight. Also, in Figure 4, it doesn't look like any parameterization was significantly superior to others.

(Response RC2-9): It was not our intention to identify a single "best" parameter set. Rather, we wanted to find what, if any, range of parameter values would lead to simulations that are consistent with the observations. We approached the analysis being open to the possibilities that many, none, or one parameter set would be consistent with observations. In our original analysis, we now think that our analysis was weakened by the fact that we considered only seven parameter sets. To better address the problem that you mentioned as well as comments from other reviewers, we carried out more simulations on more densely sampled parameter space and changed analysis method to compare the differences between simulations and observations. Therefore, Figure 4 will not be in our revised manuscript. More details are presented in Response RC1-9.

Having now carried out these additional simulations, we think it is an interesting result that the only parameterizations that were consistent with observed leaf, wood, and fine root productivity all had a positive relationship between relative allocation to fine roots and soil P concentration. Such a parameterization had not been applied in previous models. We do not yet have the data to know if these "pos" parameterizations will be better than "neg" or "const" parameterizations on timescales longer than three years. Thus, we think that this result motivates further field and modeling work (see also in Response CC1-12).

**References**

Braghiere, R.K., Fisher, J.B., Allen, K., Brzostek, E., Shi, M., Yang, X., Ricciuto, D.M., Fisher, R.A., Zhu, Q. and Phillips, R.P., 2022. Modeling global carbon costs of plant nitrogen and phosphorus acquisition. Journal of Advances in modeling earth systems, 14(8), 2022MS003204.

Friedlingstein, P., Joel, G., Field, C. B. and Fung, I. Y., 1999. Toward an allocation scheme for global terrestrial carbon models, Global Change Biol., 5, 755-770.

Medvigy, D., Wang, G., Zhu, Q., Riley, W.J., Trierweiler, A.M., Waring, B.G., Xu, X. and Powers, J.S., 2019. Observed variation in soil properties can drive large variation in modelled forest functioning and composition during tropical forest secondary succession. *New Phytologist*, *223*(4), pp.1820-1833.

Xu, X., Medvigy, D., Powers, J.S., Becknell, J.M. and Guan, K., 2016. Diversity in plant hydraulic traits explains seasonal and inter-annual variations of vegetation dynamics in seasonally dry tropical forests. *New Phytologist*, *212*(1), pp.80-95.

Zhu, Q., Riley, W.J., Tang, J., Collier, N., Hoffman, F.M., Yang, X. and Bisht, G., 2019. Representing nitrogen, phosphorus, and carbon interactions in the E3SM land model: Development and global benchmarking. Journal of Advances in Modeling Earth Systems, 11(7), 2238-2258.

---

## Author Response (AR1)

Dear Editor,

Thank you for the opportunity to revise our manuscript. We have made changes following the reviewer and community comments, and we think that the manuscript is significantly improved as a result. We have labelled all of our responses for ease in cross-referencing. Responses to Reviewer 1 are labelled "RC1", responses to Reviewer 2 are labelled "RC2", and responses to the community comment are labelled "CC1". Line numbers cited here refer to the clean, revised version and not the tracked-changes version of the manuscript. Reviewer and community comments are presented in blue font; our labeled responses are in ordinary black font.

**Response to Reviewer 1**

**General comments**

This study integrated a representation of phosphorus-dependent relative allocation to root tissues into the ED2 model. The model was simulated at a site in a tropical dry forest in Costa Rica for +N, +P, and +NP fertilization treatments. Modelled results were compared to empirical observations. The model was then simulated over 30 years to examine the influence of the new process representation over a longer time scale. The comparison between a model and empirical observations of experimental manipulations of nutrient input is very useful for model development. However, it was unclear whether the process that was represented in this study (increasing allocation to fine roots with increasing soil P) is prevalent in systems outside of this site, what its underlying mechanisms are, and how it relates to other central processes (such as the relationships between allocation to fine roots and soil nitrogen or water). Furthermore, the statistics used to establish the results were unclear.

(Response RC1-1): Thank you for your review. We are happy that you found the comparison helpful and are very grateful for your constructive feedback. In our responses below, we have addressed the points of clarification that you raised. We also carried out a new statistical analysis, largely following your advice. We think that incorporation of these changes have resulted in a greatly improved manuscript.

**Specific comments**

1. The premise of this study needs to be better established. What is the mechanism underlying increasing allocation to fine roots with increasing soil P?

(Response RC1-2): This study investigated the consequences of three related premises: (a) plants increase fine root production with increasing soil P, (b) plants decrease fine root production with increasing soil P, and (c) plant fine root production is independent of soil P (see lines 85-88). Regarding (a), several studies have shown that roots proliferate in nutrient-rich patches (Pregitzer et al. 1993; Robinson 1994; Zhang and Forde 1998; Robinson et al. 1999; Fransen et al. 1999; Hodge et al. 1999; Jing et al. 2010; Li et al. 2012). This proliferation may be related to enhanced root morphological plasticity with increasing soil P, which increases nutrient uptake per unit construction cost (Fitter 1994; Eissenstat and Yanai 2002; Zhang et al. 2016). It may also be physiologically adaptive if it allows for faster uptake of nutrients from the soil (Jackson et al. 1990; Hodge et al. 2004). It is possible that plants would also overallocate to fine roots to

better compete with neighbors (Gersani et al., 2001; Zea-Cabrera et al., 2006; Farrior et al., 2013). See lines 40-44, 58-64, 372-384. We discussed premise (b) in the original manuscript. Regarding premise (c), it could be that fine root production depends more on water or N than on P. In this case, relative allocation to fine roots would not be directly sensitive to soil P. See our response RC1-5 below for more on this point.

*Does this response occur in other ecosystems or are the only observations from the Costa Rica site? The introduction gave several examples of how different ecosystems and different individuals within a given ecosystem respond differently to N and P fertilization. Are there any patterns that emerge across ecosystems? If this response is specific to a single or a small number of sites, why should it be represented in TBMs?*

(Response RC1-3): Our observations are only from the Costa Rica site. Cunha et al. (2022) carried out a fertilization experiment on old-growth Amazonian sites and also found that relative and absolute allocation to fine roots increased under P fertilization. In another Amazonian fertilization experiment, Lugli et al. (2021) reported that absolute fine root production increased, but did not comment on relative allocation to fine roots. See lines 51-53, 57-59, 65-70. Given these studies, we think that this response is relevant to TBMs.

Previous global meta-analyses have considered absolute production of fine roots (Yuan and Chen 2012) and relative production of fine roots (Li et al. 2016). Globally, these agree that absolute production of fine roots increases with P fertilization. Li et al. (2016) found that relative production of fine roots declined. But it should be noted that the number of tropical studies included had large variability and methodological issues have been pointed out with respect to the tropical sites (Wright 2019).

*Have empirical studies indicated that this is important for larger C fluxes? This is somewhat touched on in the Discussion but its prevalence was not clear.*

(Response RC1-4): We are not aware of its relevance for larger C fluxes on the basis of previous empirical studies. Also, it is difficult to experimentally manipulate relative allocation to ascertain its importance for larger C fluxes. However, based on our modeling, we do show a strong sensitivity of aboveground biomass to parameterization, so we expect that other C fluxes will also be sensitive. Part of our intention is to use the modeling results to stimulate hypotheses and prompt more field work in the future (lines 341-342, 465-468).

2. *How do other factors interact to determine relative allocation to roots? Water and nitrogen should play important roles as well. Is it valid to only focus on P (especially given that the results suggest that it is increased water uptake that seemed to drive AGB)?*

(Response RC1-5): We focused the relationship between soil P and relative allocation to fine roots on the basis of a previous fertilization experiment (Waring et al. 2019). That experiment showed that relative allocation to fine roots was impacted by P fertilization but not N fertilization. Furthermore, the experiment spanned years of relatively high and low rainfall, and thus allowed us to see how the actual fine root-to-leaf productivity ratio varied with rainfall. Specifically, rainfall in 2015 was 628 mm, rainfall in 2016 was 1754 mm, and rainfall in 2017 was 2050 mm. The observed leaf:fine root productivity ratio did not exhibit a clear relationship

with rainfall. Averaged across treatments, this productivity ratio was 0.37 in 2015, 0.44 in 2016, and 0.24 in 2017. We also looked at the leaf:fine root productivity ratio by treatment. In all cases, this productivity ratio was intermediate in 2015, largest in 2016, and smallest in 2017. Because the observations lacked a clear correlation with rainfall, we decided not to focus on the impacts of water on the leaf:fine root productivity ratio. A discussion of this rationale has been incorporated into the methods section of our manuscript (lines 119-125, 172-174).

We acknowledge that our original manuscript did draw a correlation between simulated transpiration and simulated aboveground biomass (AGB) in our 30-year sensitivity analysis. However, prompted by this comment as well as a comment from a community reviewer (response CC1-11), we carried out more detailed analysis of this point. After this further analysis, we do not think that the correlation between transpiration and AGB was causal. Instead, the higher transpiration in the "pos" schemes occurred because the "pos" schemes had larger leaf area index. In our revision, Fig. 5 shows the leaf area index. It is larger under certain parameterizations because of both the *r2l* parameter and because of a compounding effect in which simulations with large leaf area index are more productive and therefore produce even more leaf area.

Additionally, I would assume that the role of other plant mechanisms to increase P uptake would be important as well, such as phosphatase synthesis and arbuscular mycorrhizae. These are likely intricately linked to fine root biomass in real ecosystems. While these do not necessarily need to be examined or modelled, they should be at least recognized in the experimental setup and discussion.

(Response RC1-6) We agree that other plant mechanisms to increase P uptake could be important as well. In the revision, we discussed additional plant mechanisms to increase P uptake, including phosphatase synthesis (Liu et al. 2015; Kong et al. 2016; Lugli et al. 2020) and arbuscular mycorrhizae (Hodge 2004; Comas et al. 2014; Eissenstat et al. 2015; Liu et al. 2015; Kong et al. 2016; Ma et al. 2018). For example, root phosphatases hydrolyze organic P-containing compounds and releasing inorganic P that is absorbable by roots; mycorrhizal associations are even more effective by enlarging the root absorbing surface per unit cost; both mechanisms provide additional P sources. Plants adaptively adjust their traits or metabolic processes in terms of effective P acquisition (Raven et al. 2018; Han et al. 2021; Aoyagi et al. 2022), and diverse P acquisition strategies are being evaluated from observations (Reichert et al. 2022).

Additionally, flexible stoichiometry could be important. How have other models approached these phenomena?

(Response RC1-7): Current models approach stoichiometry differently. Some have fixed stoichiometries (e.g. JSBACH, Goll et al., 2012; CLM-CNP, Yang et al., 2014; JULES-CNP, Nakhavali et al., 2022). Some account for stoichiometric flexibility by prescribing ranges for each pool based on empirical studies (e.g. CASACNP, Wang et al., 2010; ORCHIDEE-CNP, Goll et al., 2017; QUINCY, Thum et al., 2019). ED2 has fixed stoichiometries in structural pools but non-fixed stoichiometries in non-structural pools. Models adjusting relative allocation of new

growth to fine roots mostly apply an idea that new growth is scaled by minimum of N and P stress scaling factor, rendering increased fine root production when P demand exceeds supply (for example, when the ratio of nonstructural C:P greatly exceeds the ratio present in plant tissues). By contrast, we parameterized the model only from the perspective of supply (soil P). We have included a discussion of these points in our revision (lines 445-448).

3. It was not made clear which PFT was being studied in these experiments. Were there multiple PFTs? Given that this is a dry tropical forest, do deciduousness and phenology play a role here? How could these results differ between tropical dry forests and tropical moist forests? Have similar experiments been conducted in tropical moist forests?

(Response RC1-8): The model included eight PFTs. Species are assigned to a PFT on the basis of three traits: wood density, specific leaf area, and legume/non-legume status. The binning of species into PFTs is discussed in Xu et al. (2016) and Medvigy et al. (2019). Deciduousness is not pre-assigned, but is rather an emergent model outcome (Xu et al. 2016). In this dry tropical forest, intra-annual variability of precipitation does influence plant phenology: new leaves are produced in April and May and shed between January and March; stems did not grow during the dry season (Waring et al., 2019). Therefore, the fertilization was conducted only during wet season; measurements of leaf/wood/fine root production were conducted differently to accommodate seasonality (Waring et al., 2019). Our model simulations and the calculation of productivity were in accordance with this experiment. Analogous experiments have been conducted in tropical moist forests (please see response RC1-3). We provided clarification and discussion of these points in the revision (lines 101-103, 106, 112-114, 135-136, 144).

4. Using different statistical analyses for leaf, wood, and root due to patterns that emerged from the observation-based data may not be the best approach. It would be a more direct comparison to use the same statistical analyses for each tissue because the biases in the empirical observations may not be present in the model outputs. Figure 4 is a central figure but it is unclear whether it shows only the control treatment or an average across treatments. Regardless, this analysis should be conducted for each fertilization treatment independently given that the premise of the study is that fertilization treatment influences relative allocation. Furthermore, are the temporal trends important here given that the same amount of fertilizer was applied each year and the experiment was only 3 years long? Given that the primary focus is the difference between tissues rather than the difference between years, it may make more sense to aggregate across years for each tissue / treatment.

(Response RC1-9): We thank the reviewer for these suggestions. We re-analyzed these results largely following this advice. First, we averaged over years. The main drawback to this averaging is that we lose some statistical power because all t-tests would need to be done on only four replicates. However, when we carried out this analysis, we found that the impact on our results was small (see below). We also performed statistical tests by treatment. An advantage of following this approach is that we no longer had concerns about mixing samples that were drawn from different distributions. We therefore performed t-tests for leaf, wood, and fine root productivity. We no longer found it necessary to do the regression on fine root productivity. As a result, productivity of leaves, wood and fine roots were analyzed in exactly the same way.

First we looked at leaf productivity. We carried out t-tests to determine whether there were significant differences between simulations and observations. In response to a community comment (please see response CC1-9), we increased the number of parameterizations that we tested. If, for a particular treatment, the p value was less than 0.05 (the threshold which we took to indicate a significant different between the simulations and the observations), we indicated that in the table below. "C" means the control plots were significantly different, "N" means the +N plots were significantly different, "P" means the +P plots were significantly different, and "B" indicates that the +NP plots were significantly different. If an entry is empty, it means that no significant differences were found for any treatment. Here are our results for leaf productivity:

| LEAF PRODUCTIVITY | a=0 | a=0.1 | a=0.2 | a=0.3 | a=0.4 | a=0.5 | a=0.6 | a=0.7 | a=0.8 |
|---|---|---|---|---|---|---|---|---|---|
| b = -60 | | | | | | | | | |
| b = -40 | | | | | | | | | |
| b = -20 | | | | | | | | | |
| b = 0 | | | | | | | | | |
| b = 20 | | | | | | | | | |
| b = 40 | | | | | | | | | |
| b = 60 | P | P | | | | | | | |

As in our original manuscript, almost all model parameterizations successfully predicted leaf productivity.

We then looked at wood productivity:

| WOOD PRODUCTIVITY | a=0 | a=0.1 | a=0.2 | a=0.3 | a=0.4 | a=0.5 | a=0.6 | a=0.7 | a=0.8 |
|---|---|---|---|---|---|---|---|---|---|
| b = -60 | P | P | P | P | P | | | | |
| b = -40 | P | P | P | P | P | | | | |
| b = -20 | P | P | P | P | P | P | | | |
| b = 0 | P | P | P | P | P | P | P | | |
| b = 20 | P | P | P | | | | | | |
| b = 40 | | | | | | | | | |
| b = 60 | | | | | | | | | |

Many parameterizations, especially with $b < 0$, $b = 0$, or relatively small $a$, did not predict wood productivity in the +P treatment. Parameterizations with $b > 0$ were mostly successful in all treatments. Again, this result is consistent with our original manuscript. However, here we have more information as compared to our original manuscript: we see that the model only ever had trouble with the +P treatment, and not the +NP treatment.

Finally, we looked at fine root productivity:

| FINE ROOT PRODUCTIVITY | a=0 | a=0.1 | a=0.2 | a=0.3 | a=0.4 | a=0.5 | a=0.6 | a=0.7 | a=0.8 |
|---|---|---|---|---|---|---|---|---|---|
| b = -60 | C,N,P,B | C,N,P,B | C,N,P,B | C,N,P,B | C,N,P,B | P,B | P,B | C,P | C,P |
| b = -40 | C,N,P,B | C,N,P,B | C,N,P,B | C,N,P,B | P,B | P,B | P,B | C,P | C |
| b = -20 | C,N,P,B | C,N,P,B | C,N,P,B | C,N,P,B | P,B | P,B | P | C | C |
| b = 0 | C,N,P,B | C,N,P,B | C,N,P,B | C,N,P,B | P,B | P | P | C | C |
| b = 20 | C,N,P,B | C,N,P,B | C,N,P | N,P | | | C | C | C,B |
| b = 40 | C,N,P | C,N | N | | | C,B | C,P,B | C,P,B | C,N,P,B |
| b = 60 | C,N | N | B | P,B | C,P,B | C,P,B | C,P,B | C,P,B | C,N,P,B |

Only four parameterizations, all with $b = 20$ or $b = 40$, successfully predicted fine root productivity in all treatments. These four parameterizations also successfully predicted leaf and wood productivity in all treatments. These new results and tables are included in the revision (lines 196-197, 264-269).

Again, the drawback of this approach is that is reduced the number of samples: each t-test was done with only four replicates. However, our new results are similar to what we reported in the original manuscript. As an additional test, we tried aggregating control and +N, and comparing that to the aggregation of +P and +NP. This procedure doubles the number of replicates. It gave very similar results (not reported here). (see lines 269-272)

5. Is a 2 year spinup sufficient? Shouldn't the spinup be run until an equilibrium is established?

(Response RC1-10): We did not want to spin up the model to equilibrium because the actual forest is only about 30 years old and is not in equilibrium. Rather, we used the observed stand structure and composition and observed soil nutrient status to initialize the model. Our two-year spin-up was done to initialize soil water. For further rationale, see Xu et al. (2016).

Technical correction

Line 81 "While models have rarely be validated on these time scales" I would argue that models are often evaluated over the past several decades (1960s to present).

Here we wanted to emphasize that nutrient fertilization experiments have usually lasted for a few years and hence the lack of long-term benchmarks for model validation. We rephrased the text.

Production units should be kg m-2 yr-1.

Thanks for pointing this out. We corrected the units.

Table 3 is challenging to interpret. Could this be transformed into a figure?

We created a simplified version of the table that we think better summarizes the main result.

Include other parameterizations in Figure 5 (additional panels).

As discussed above, this figure was eliminated in the revision.

Include other treatments in Figure 8 (additional panels).

As discussed above, this figure was eliminated in the revision.

Figure 6: Clarify if this is averaged across treatments or if this is the control treatment only.

Done.

**Response to Reviewer 2**

Shuyue Li et al., conducted a modeling study to investigate how plant allocation in response to nutrient fertilization. They used nutrient enabled ED2 model with various different parameterizations on biomass allocation under control and fertilized conditions over tropical dry forest. Data from an fertilization experiment at Costa Rica forest was used for model comparison and validation. The paper is well organized and presentation is smooth. Below I have a few suggestions and comments.

(Response RC2-1): Thank you for your review and helpful suggestions and comments. Please see our responses to each of them below.

1. introduction, first paragraph needs to be improved, Nutrient availability could affect plant activity in many different ways. The most relevant (to this paper) way is through mediation C/N/P allocation and biomass construction. However, the first paragraph try to explain how nutrient availability could affect plant response to CO2 enrichment, which is not much relevant here.

(Response RC2-2): In the first paragraph of the manuscript, we wanted to provide the overall context for this study: more process-level understanding of nutrient limitation is essential for reliable prediction on the primary production of terrestrial ecosystem under future environmental change. Yet we think that this is a fair critique. We revised the first paragraph to be briefer on the general context and to better introduce the idea that nutrient limitation can affect relative allocation, with consequences for production (lines 32-36).

2. introduction, paragraph 2 and 3 provide a nice summary of many fertilization experiments for tropical trees. However, each fertilization experiment was discussed individually. I would suggest adding some discussion about why and how experiment results differ from one another to improve the coherence of the summary.

(Response RC2-3): Following this suggestion, we have revised paragraphs 2 and 3. We now begin the discussion by describing results from a few meta-analyses, which we hope will improve the coherence. Then, we discuss a few individual studies that were published after the meta-analyses. We have discussed a few reasons for differences the studies (lines 37-64).

3. introduction, paragraph 5 and 6 highlight the need to investigate and improve the allocation scheme under long-term fertilization for current generation CNP models. In this case, a survey of allocation schemes used by current generation models are necessary, for example some models assume constant allocation, some assume multiple resource coordination, some are based on carbon cost …Besides the seven CNP models mentioned in this section, two more recent global CNP models are:FUN-CNP: Braghiere, R.K., Fisher, J.B., Allen, K., Brzostek, E., Shi, M., Yang, X., Ricciuto, D.M., Fisher, R.A., Zhu, Q. and Phillips, R.P., 2022. Modeling global carbon costs of plant nitrogen and phosphorus acquisition. Journal of Advances in modeling earth systems, 14(8), 2022MS003204. ELM-CNP: Zhu, Q., Riley, W.J., Tang, J., Collier, N., Hoffman, F.M., Yang, X. and Bisht, G., 2019. Representing nitrogen, phosphorus, and carbon interactions in the E3SM land model: Development and global benchmarking. Journal of Advances in Modeling Earth Systems, 11(7), 2238-2258.

(Response RC2-4): We thank the reviewer for bringing these additional models to our attention, and we now mentioned them in the text. We also provided further discussion on the allocation schemes of current generation CNP models, but we thought that the best place for that was in the Discussion (see section 4.4) rather than the Introduction. See lines 70-75, 434-448.

4. section 2.3, r2l is a function of soil P concentration (psol), I wonder mathematically will this equation lead to huge variability of r2l parameters especially at the time when fertilizers were applied. Maybe showing a figure of r2l during the 3 years of fertilization experiment will help to clarify this.

(Response RC2-5): The variability of the r2l parameter depends on treatment and parameterization. Without P fertilization, the variability in r2l is relatively small. The variability is largest under P fertilization with the "pos" parameterizations, where r2l ranges mostly from 0.4 to 1.0. It is possible that this magnitude of variability is realistic. We have included additional discussion (lines 249-253) and a figure in the manuscript.

[Figure]

5. section 2.4.1. Vegetation and soil are both initialized with in situ observations, rather than being determined by long-term spinup. Such approach often time will result in an dis-equilibrate vegetation and soil processes. Therefore, after initialization the vegetation and soil states will quickly changes towards quasi-equilibrate conditions, which could be largely different from the initialized conditions. I wonder if the re-equilibration also occur in ED2, how long does it re-equilibrate, and how that affect fertilization results?

(Response RC2-6): This question is difficult to answer because our study system is a 30-year secondary forest and the actual forest is not equilibrated yet. Because the actual forest is still unequilibrated, we did not spin up the model to equilibrium. The two year spinup was sufficient to initialize soil water (see also in Response RC1-10). Despite this short spinup, the model has been shown to simulate aboveground biomass reasonably well (Medvigy et al. 2019) and no obvious transients in leaf area index have been observed (Xu et al. 2016). The plant-available soil nutrient pools (Fig. 3) also do not exhibit much of a transient. The slowly varying soil pools were initialized from observations and were not expected to change much over the course of this study because they are slowly varying. We commented briefly on this point in the revision (lines 192-194).

6. section 2.4.1. It was mentioned that fine root production was evaluated with linear regression, however, it also mentioned linear regression was not appropriate because there existed only three years of data. Here needs more clarification.

(Response RC2-7): We changed our analysis of fine root production. Please see our detailed responses in Response RC1-9. In short, we reanalyzed our simulation results by first averaging over years, and then performing t-test for leaf, wood, and fine root productivity by treatment, to determine whether there were significant differences between simulations and observations. By doing so we no longer need linear regression to evaluate model performance on fine root production. We used this modified analysis in the revision (lines 196-197).

7. Figure 2, most of the simulated variability of NO3, NH4 already exist in control run (solid blue lines), it doesn't look like there were sudden increase of NH4 or NO3 right after the N fertilization. Also, it will be helpful, if the fertilization date could be marked on the x-axis.

(Response RC2-8): We agree with this comment. Note that the observations in Waring et al. (2019) also did not show any step changes in NO3 or NH4 in response to fertilization. In the model, this is because leaching is an important pathway for N loss, especially during wet seasons when fertilizers were applied. We modified Fig. 3 to display approximate fertilization time.

8. Section 4.1. It's still not clear to me which parametrization is the best. It was stated that "only one of the 13 parameterizations that we tested was able to simultaneously simulate leaf, wood and fine root (missing word) production consistent with the observations". Here, the screensful parameterization needs to be highlight. Also, in Figure 4, it doesn't look like any parameterization was significantly superior to others.

(Response RC2-9): It was not our intention to identify a single "best" parameter set. Rather, we wanted to find what, if any, range of parameter values would lead to simulations that are consistent with the observations. We approached the analysis being open to the possibilities that many, none, or one parameter set would be consistent with observations. We modified the text of the Introduction to make this point clearer. See lines 85-90, 233-234.

To better address the problem that you mentioned as well as comments from other reviewers, we carried out more simulations on more densely sampled parameter space and changed analysis method to compare the differences between simulations and observations. Therefore, the original manuscript's Figure 4 has been removed. More details are presented in Response RC1-9.

Having now carried out these additional simulations, we think it is an interesting result that the only parameterizations that were consistent with observed leaf, wood, and fine root productivity all had a positive relationship between relative allocation to fine roots and soil P concentration (lines 335-337). Such a parameterization had not been applied in previous models. We do not yet have the data to know if these "pos" parameterizations will be better than "neg" or "const" parameterizations on timescales longer than three years. Thus, we think that this result motivates further field and modeling work (see also in Response CC1-12) (lines 463-464).

**Response to Community Comment 1**

The authors confront a dynamic vegetation model with experimental data from a nutrient fertilization experiment in Costa Rica and assess the validity and consequences of alternative root allocation parameterization. The overarching question the authors address is how root allocation responds to changes in nutrient availability, which is a very valid question to pursue. Evaluating models and their underlying assumptions with direct field observations and experiments is necessary to advance our understanding and gain confidence in model predictions. I appreciate the effort the authors have undertaken with the demographic model ED2, and their joint effort with experimental findings and expertise to advance root dynamics and their interactions with nutrient availability. However, I currently cannot recommend publishing the manuscript due to some concerns.

(Response CC1-1) Thank you for these helpful comments, and for the many great suggestions that follow. In a revision, we think that we would revise how we state our overarching questions in order to better align them with our results and discussion. Specifically, we are trying to answer: (a) What model assumptions enable or prevent the model from being consistent with the observations? (b) How can we use the model to generate hypotheses on the longer-term consequences of allocation assumptions? See lines 83-84, 93-94.

Main concern:

1. The ecological underpinnings of the chosen parameter sets in regard to fine root allocation are not sufficiently clear. This includes the ecological theory to justify choosing the parameter sets, the discussion of the consequences of these parameter sets, and placing the chosen parameter set and findings within the literature and previous modeling efforts.

(Response CC1-2) Thank you for this comment. Reviewer 1 brought up similar concerns. Please see our responses RC1-2, RC1-3, and RC1-7.

2. The study covers short-term and long-term process effects of the different parameterizations, whereby we can expect different outcomes. The experiment finds that fine root production increases with fertilization.

(Response CC1-3) We agree with this comment.

That does not necessarily mean that higher fertile sites will be characterized by higher fine root production.

(Response CC1-4) We are open to the idea that allocation responses to P fertilization on a three-year time scale might differ from correlation analysis (e.g., correlation of soil P with fine root productivity across a strong fertility gradient). Individuals growing on naturally high-P sites, for example, may have had sufficient time to acclimate to their local environment. Also, on decadal to multidecadal time scales, forest demand for P may change as a function of forest age or as species turn over, and such changes can affect relative allocation. We included additional discussion on the distinction between fertilization experiments and correlation analysis in a revision (lines 357-358, 369-371, 423-424).

The others touch on these aspects of timing in the discussion, and the significance of this difference is crucial for the study, however insufficient emphasis is placed on this in working out the premise of the study and discussing the findings.

(Response CC1-5) We are not sure that we fully understood this comment. Which "others" are being referred to here? We are very open to the idea that short term (3-year) and long term (30-year) responses might be different, and that is indeed part of the reason why we carried out 30-year simulations.

We note that in our 30-year model simulations, successional changes in forest structure, composition, and function are allowed to occur. But the allocation rule remains fixed (no acclimation). We think it is interesting that the parameterizations that matched the 3-year observations most closely simulated the least AGB accumulation on 30-year time scales with P fertilization. This model result does beg the hypothesis that some acclimation would occur on 30 year time scales. This idea can mathematically be incorporated into the model by having relative fine root allocation be a saturating function of soil P rather than a linear function. Such a rule would help to prevent over-allocation to fine roots under the very high soil P concentrations associated with P fertilization. We hope that these remarks speak to your concern and we included additional discussion on these points in our revision (lines 423-432).

Associated with this, the experiment takes place in a young forest stand, the implications of this deserve more discussion.

(Response CC1-6) We agree that forest age could be an important variable related to relative allocation to fine roots. Forests of different ages would be expected to have different composition, structure, and demographic rates. It is interesting that the pattern seen in the relatively young forest of our analysis was similar to the pattern seen in the old growth forests studied by Cunha et al. (2022). In a revision, we included additional comment on this point (lines 80-81).

3. Direct nutrient acquisition via fine roots is only one of several possible mechanisms of how plants can acquire nutrients. The authors mention that the trees are associated with arbuscular mycorrhizae but the implications of this in regard to the outcome of the experiment, and how this might have affected the model performance are not addressed.

(Response CC1-7) Please see our response RC1-7. Waring et al. (2019) found root colonization by mycorrhizal fungi did not vary among nutrient treatments (lines 125-126). In the manuscript we discussed the possibility of increased root production but reduced allocation to mycorrhizae, which together might show overall decreased belowground allocation.

Additional comments methodology:

The chosen parameter sets reflect negative, constant, and positive relationships between fine root production and soil P. The negative one would reflect the resource-dependent parameterization, however not via internal plant demand and supply (as commonly done) but depending on external P supply, in a linear fashion. This is quite different from any of the previous model approaches, and the

parameterizations are all based on a linear relationship between soluble soil P and root allocation. A discussion of the ecological underpinning of this model and the parameterization approach would be helpful. The authors touch upon the ecological theory they are addressing only in the discussion part.

(Response CC1-8) Please see our response CC1-2.

Parameters a and b are coordinated to yield a similar 0.3 root-to-shoot ratio in control plots, but to different settings in fertilized plots. The reason for not testing different parameterization settings in the control plot was not clear to me, it would be helpful if the authors could elaborate on this.

(Response CC1-9) Thank you for this particularly helpful comment. We carried out more simulations so that we more densely sampled parameter space. Some of the main results are presented in our responses RC1-5 (30-year time scale) and RC1-9 (3-year validation).

The allocation process as a whole in ED2 is not sufficiently explained. The calculation of the daily leaf and root allocation, and the allometric equations to determine maximum leaf and root allocation, need to be described and included in the discussion. Similarly, the PFTs that are modeled and their parameterizations are not described.

(Response CC1-10) We originally did not present this information because it can be found in other publications (Longo et al. 2019a, Medvigy et al. 2019). However, we included more details in our revision (lines 149-154).

Additional comments results:

One positive parameterization was the only one that agreed well with experimental observations over the 3-year period. Positive parameterizations led to the highest AGB over 30 years, however, not due to alleviation of nutrient limitation but water stress. It would be helpful if the reader would understand how that occurred process-wise. What was the actual allocation to fine roots? And why did that not alleviate nutrient stress?

(Response CC1-11) For a process-level explanation, see response RC1-5. The actual allocation to fine roots is shown in Figure 5 of the revised manuscript.

Similarly, with P fertilization, positive parameterizations led to too many fine roots over the 30-year time scale. As is described in the abstract as the main finding. Does that not indicate that the chosen parameterization works well for the short term but, importantly, not well in the long term? Is such an assumption about root allocation advisable then? It would be helpful if the authors go beyond "experiments need to measure leaf, root and wood production" in their main take-away for the paper, and rather elaborate on recommendations for model development.

(Response CC1-12) We think that the parameterization works well in the short term in the sense that it yields simulation results that are consistent with observations. We do not know how well the parameterization would work in the long term because we do not have long-term observations to compare the simulations to. Thus, we see a strong need for longer-term experiments exactly in order to make such assessments. We did not mean to imply that the positive parameterization is a "chosen" parameterization, and all of our results (except the

baseline) displayed multiple parameterizations. We can try to be more clear about this point in a revision (line 443-444).

We offer several additional thoughts spurred by this comment, which we have now worked into our Discussion: (1) It could be that a saturating response of relative fine root allocation to soil P (rather than a linear response) would maintain the model's capability of simulating the 3-year observations and also maximize AGB under fertilized and unfertilized conditions. But this would be a more complicated parameterization than the one we employed, and so we did not implement it. (2) It could be that P fertilization over 30 years would expose trees to soil P concentrations that are well outside the natural range. In such a novel environment, the response may well be maladaptive. (3) Under "normal" unfertilized conditions, there does not appear to be any problem with positive parameterizations (Fig. 6a). It could also be that increasing the maximum size of the nonstructural P pool would increase AGB accumulation under the positive parameterizations. See lines 423-432.

We think that model development and experimental work should be coupled. Given our modeling results, we think that the natural next step should be more experiments. Additionally, testing of different parameterizations in different locations like the Amazonian sites. Model intercomparison may also be helpful (lines 465-468).

Given that the experiment took place in a tropical dry forest, the anticipated interactions between soil fertility and water stress are intriguing and it would be helpful if the authors elaborate on the alleviation of water stress in the 30-year simulation that has been touched upon.

(Response CC1-13) Please see our response RC1-5.

Maybe the agreement at the 3-year time scale is not sufficient to evaluate.

(Response CC1-14) We are not exactly sure what is meant here. A three-year time scale is not uncommon in ecosystem experiments. Further, the model represents processes on time scales ranging from seconds to centuries. Ideally, we would constrain the model over a range of time scales (for example, see Trugman et al. 2016; Longo et al. 2019b). Because the Waring et al. (2019)'s fertilization occurs within the range of time scales that the model simulates, we do think it is appropriate to validate the model on a three year time scale. Note that the model has been validated on somewhat longer time scales at nearby unfertilized plots (Medvigy et al. 2019).

Can the control plots not also be used to evaluate the parameterization, representing rather long-term dynamics? Should the parametrizations not be tested on the control and experimental plots, equally?

(Response CC1-15) We agree that this would be helpful. Please see our response RC1-9. These new results have been included in the revision.

The authors state that ED2 has been validated before. Did the constant parameterization work well before? Did ED2 employ a resource-dependent allocation scheme before? It would be helpful to outline which allocation approaches agreed well with observations and which did not in previous studies with ED2.

(Response CC1-16) ED2 has been validated at a range of ecosystems and for a range of quantities (Medvigy et al. 2009; Medvigy and Moorcroft 2012; Trugman et al. 2016; Xu et al. 2016; Levy-Varon et al. 2019; Longo et al. 2019b; Medvigy et al. 2019). The "constant parameterization" is the one built in to the basic model code, and has never been contrasted with a resource-dependent allocation scheme before. Generally speaking, the model has only been validated for aboveground quantities like AGB production, leaf production, and eddy fluxes. We show here that many parameterizations lead to reasonable wood and leaf production (as the model has been previously validated for), but few lead to reasonable fine root production (as the model has rarely or never been validated before). See lines 130-132, 170-172, 348-352.

The resource limitation theory postulates plant allocation is adjusted to acquire the most limiting resource. Many ecosystem models to date, adopt a resource-dependent allocation scheme, so they would predict that more roots would be produced if soil nutrients were limited, irrespective of the return. The authors find that the experimental results contradict this hypothesis. I believe this apparent contradiction might be a time issue, so that roots grow to acquire the soluble P, while root allocation would decrease once P limitation is alleviated. It might also have to do with mycorrhizal interactions, previous outsourcing of phosphorus acquisition to mycorrhizae now becomes less beneficial with increased nutrient supply and plants switch to "do-it-yourself". Even if we cannot be certain about any of these hypotheses, a discussion thereof would be helpful.

(Response CC1-17) Thank you for these points. We think that they have largely been addressed in our author comments (different ecosystem models were mentioned in RC1-7, timing issues in response CC1-5 and CC1-12, and mycorrhizal colonizations in CC1-7). We have revised our manuscript to include discussion of these points.

Line comments:

1. 336 please specify: allocation parameterization sensitive to external nutrient availability.

(Response CC1-18) This paragraph has been significantly edited.

2. 383- 341 what is the implication of that finding?

(Response CC1-19) Is there a typo in the line numbers specified here? We were not sure which lines the reviewer referred to.

3. 344 the model is microbial-explicit. That is an important aspect of the model. It would be helpful if this is included in the manuscript and discussed.

(Response CC1-20) We added a bit more on this in the model description (lines 155-158).

4. 365 For future efforts, the authors can consider evaluating the model and different parameter sets at different locations in the tropical biome, and including an evaluation along a soil fertility gradient, as well as experimental changes. The combination of both would be helpful to discern short-term and long-term effects.

(Response CC1-21) Thanks for the suggestion and please see also our response CC1-4. We added this to the discussion.

5. 369 The experiment near Manaus found increased primary productivity in response to fertilization, indicating that production was limited by phosphorus. To my understanding, this finding is still consistent with the resource limitation hypothesis, since plants allocated carbon to roots to acquire the nutrients that they were in demand of.

(Response CC1-22) Of concern to us is that relative – and not just absolute – allocation to fine root increased in Cunha et al. (2022) in response to P fertilization (lines 51-52).

6. 375 As the authors note, roots are there for acquiring multiple resources at once. A discussion of these interactions would be helpful. The analogy to rain roots is interesting, the authors could elaborate here. The root production after fertilization could potentially be a similar short-term effect to acquire the limiting resource.

(Response CC1-23) We could briefly elaborated on these points. Also relevant are our responses CC1-4, CC1-5, and CC1-12.

7. 379 The paragraph on field observation is helpful, however, root stock and root production seem to be mixed up. Since the model evaluation deals with root production, the authors could elaborate on those aspects of the measurements.

(Response CC1-24) I'm not sure that we mixed up root stock and root production, but we should have also mentioned root production. This has now been added (lines 397-398).

8. 387 It would be helpful if the authors elaborate on the interactions with mycorrhizae, here, and the implications for this study.

(Response CC1-25) We commented on this in a revision. Please see response CC1-7.

9. 389 The "supply-limited" hypothesis, which is the basis of this study, should be introduced earlier and placed in context with the alternative hypotheses.

(Response CC1-26) Thanks for the suggestion. We introduced this concept earlier in the Introduction.

10. 390 This section could benefit if it discussed what has been learned from this and the previous approaches.

(Response CC1-27) We agree that more can be said here. Some important points include: (1) the sensitivity on one time scale (3 years) can be different from the sensitivity on another time scale (30 years). (2) Different measures of productivity differ in their sensitivity. Just because one measure of productivity is sensitive to a particular parameter does not mean that the model is generally sensitive to that parameter. (3) For some measures of productivity, model sensitivity strongly depends on the nutrient context (that is, whether or not P fertilization is applied). See lines 405-413.

11. 398 Similarly, in this section, it would be helpful if the authors go beyond the summary of the results here and discuss the implications of these findings. See the main comments above.

(Response CC1-28) Thanks for the suggestion. We disagree with the characterization of this text as a "summary of results"; rather, the text is more of an "interpretation" of results, and so it

should be appropriate for a discussion section. The text discusses implications in other places, especially in Section 5, but also to a certain extent in Section 4.4 and other locations.

> 12. 424 Here it is crucial to discuss that resource-dependent model approaches are not exactly comparable to the negative parameterization applied in this study. See the main comments above.

(Response CC1-29) We have revised the text to make this clearer (lines 437-444).

**References**

[revised manuscript text omitted]

---

## Author Response (AR2)

Dear Editor,

Thank you and the reviewers for the additional feedback on our manuscript. The reviewer lists some good points for clarification, and we have tried to address them in our latest revision. Reviewer and community comments are presented in blue font; our labeled responses are in ordinary black font. Thank you again for your consideration.

**Response to the Reviewer**

Lines 21-24: This is a little unclear and I think the more interesting result to highlight here is that: Without P fertilisation, the model is insensitive to the parameterisation, whereas, with P fertilisation, the model is highly sensitive to the parameterisation.

We agree that this is a good point and have edited the text.

Line 33: See Evaluating nitrogen cycling in terrestrial biosphere models: a disconnect between the carbon and nitrogen cycles (Kou-Giesbrecht et al. 2023).

We agree that this reference is useful and have added it.

Line 70: See Terrestrial Phosphorus Cycling: Responses to Climate Change (Menge et al. 2023).

We agree that this reference is useful and have added it.

Figure 1: Add legend that shows blue/grey colours.

Done.

Figure 4: y-axis label is very hard to read.

We increased the font size.

Line 270-271: Please add this analysis to supplement.

We added the analysis as a supplement.

Lines 340-342: This sentence is very vague.

We added an example to make it more specific.

Line 367-369: Clarify and discuss that this acclimation would occur in the real world and not the model.

OK, we added a bit more detail to the text.

Line 372: This paragraph and the following paragraph would be more useful at the beginning of the discussion section.

We respectfully disagree. We think that the Discussion's first paragraph sets a more appropriate scope for the Discussion section. While this paragraph starting on 372 describes an important question, we thought it was a bit too narrow to lead the overall discussion.

Line 395: I think the more important argument here is that models rarely represent phosphatases or mycorrhizae. Also, Line 126 says that mycorrhizal fungi did not vary which contradicts this statement.

We added text reminding the reader that models rarely represent phosphatases or mycorrhizae. We also added some clarification. It is possible that carbon allocation to mycorrhizae changed without there being a change in the percentage of colonized root length.

Line 424-427: This hypothesis is unclear.

We added some clarification.